# Controlling Underestimation Bias in Constrained Reinforcement Learning for Safe Exploration

Shiqing Gao [1]    Jiaxin Ding [1]    Luoyi Fu [1]    Xinbing Wang [1]

## Abstract

Constrained Reinforcement Learning (CRL) aims to maximize cumulative rewards while satisfying constraints. However, existing CRL algorithms often encounter significant constraint violations during training, limiting their applicability in safety-critical scenarios. In this paper, we identify the underestimation of the cost value function as a key factor contributing to these violations. To address this issue, we propose the Memory-driven Intrinsic Cost Estimation (MICE) method, which introduces intrinsic costs to mitigate underestimation and control bias to promote safer exploration. Inspired by flashbulb memory, where humans vividly recall dangerous experiences to avoid risks, MICE constructs a memory module that stores previously explored unsafe states to identify high-cost regions. The intrinsic cost is formulated as the pseudo-count of the current state visiting these risk regions. Furthermore, we propose an extrinsic-intrinsic cost value function that incorporates intrinsic costs and adopts a bias correction strategy. Using this function, we formulate an optimization objective within the trust region, along with corresponding optimization methods. Theoretically, we provide convergence guarantees for the proposed cost value function and establish the worst-case constraint violation for the MICE update. Extensive experiments demonstrate that MICE significantly reduces constraint violations while preserving policy performance comparable to baselines.

## 1. Introduction

Constrained Reinforcement Learning (CRL) has become a powerful framework for addressing decision-making tasks with safety requirements (Stooke et al., 2020; Xu et al., 2021; Yang et al., 2022; Gao et al., 2024), such as robotics (Tang et al., 2024) and autonomous driving (Zhang et al., 2024), by integrating constraints into Reinforcement Learning (RL) policy optimization. However, existing CRL methods often exhibit persistent constraint violations during policy training (Sootla et al., 2022a;b), undermining their reliability in real-world deployments.

A key factor driving constraint violations is the underestimation of the cost value. In CRL, unsafe high-cost regions pose the risk of constraint violations, prompting policy updates to reduce the cost value. However, the inherent noise in function approximation disrupts the zero-mean assumption under a minimization objective (Thrun & Schwartz, 2014), resulting in cost underestimation. Consequently, high-cost states may appear deceptively safe, enticing the agent to explore them for potentially high rewards. This underestimation leads to frequent constraint violations during training, even when an optimal safe policy theoretically exists.

Although biases in cost estimation are generally undesirable, they are not inherently detrimental and can enhance policy learning and exploration in certain scenarios (Lan et al., 2020; Karimpanal et al., 2023). Underestimation generally encourages exploration, while overestimation tends to discourage exploration. Specifically, in high-cost regions, overestimation can help prevent unsafe exploration to reduce risks, whereas underestimation may induce excessive exploration of unsafe states. Therefore, the key challenge is to shape cost estimation to mitigate harmful underestimation in high-cost regions while ensuring safe exploration without being overly conservative.

To address this challenge, we propose the Memory-driven Intrinsic Cost Estimation (MICE) algorithm, integrating extrinsic and intrinsic cost value updates to mitigate underestimation while exploiting controlled bias for safer exploration. Inspired by flashbulb memory (Conway, 2013), where humans vividly recall significant experiences to avoid risks, we equip agents with a flashbulb memory module that stores previously explored unsafe states, enabling the identification of high-cost regions. The intrinsic cost is derived from the flashbulb memory, formulated as the pseudo-count (Badia et al., 2020) of the current state visiting high-cost regions in

[1]Shanghai Jiao Tong University, Shanghai, China. Correspondence to: Jiaxin Ding <jiaxinding@sjtu.edu.cn>.

*Proceedings of the $42^{nd}$ International Conference on Machine Learning*, Vancouver, Canada. PMLR 267, 2025. Copyright 2025 by the author(s).

memory. Then we introduce an extrinsic-intrinsic cost value update that mitigates underestimation by augmenting extrinsic cost estimates in high-cost regions, with a balancing factor to correct excessive bias. Further, we propose an optimization objective within the trust region, ensuring alignment between the updated policy and the policy generating the flashbulb memory samples. We also provide corresponding solution methods for this objective. Theoretically, we establish a constraint bound for the extrinsic-intrinsic cost value function and provide a worst-case constraint violation under the MICE update, ensuring minimal constraint violations during training. Additionally, we prove convergence guarantees for the proposed cost value function. Extensive experiments demonstrate that MICE significantly reduces constraint violations while ensuring robust policy performance. Our contributions are summarized as follows:

- We introduce a novel Flashbulb Memory mechanism that records high-cost regions, enabling the derivation of an intrinsic cost to effectively mitigate the prevalent underestimation in these regions.

- We propose an Extrinsic-Intrinsic Cost Value Update that incorporates the intrinsic cost estimate and applies a bias correction strategy, ensuring more robust cost estimation and safer exploration in high-cost regions.

- We provide a theoretical analysis with convergence guarantees and constraint violation bounds, along with practical optimization methods that demonstrate significant improvements in constraint satisfaction and policy performance through extensive experiments. The code for this paper is available in `https://github.com/ShiqingGao/MICE`.

## 2. Related Work

**CRL methods.** CRL optimization methods can be classified into primal-dual and primal approaches. Primal-dual methods (Ding et al., 2021; Ying et al., 2024) convert constrained problems into unconstrained ones via dual variables. NPG-PD (Ding et al., 2020) guarantees global convergence with sublinear rates. PID Lagrangian (Stooke et al., 2020) introduces proportional and differential control to mitigate cost overshoot and oscillations. However, primal-dual approaches are sensitive to initial parameters, limiting their application (Zhang et al., 2022). In contrast, primal methods directly optimize constrained problems in the primal space (Zhang et al., 2020; Yu et al., 2022; Gao et al., 2024). CPO (Achiam et al., 2017) enforces performance and constraint violation bounds within a trust region. PCPO (Yang et al., 2020) alternates between reward improvement and policy projection into feasible regions. CUP (Yang et al., 2022) provides generalized theoretical guarantees using the generalized advantage estimator (Schulman et al., 2015). How-

ever, both primal and primal-dual methods often encounter significant constraint violations during training, primarily due to the underestimation of cost value. To address this, we propose a memory-driven intrinsic cost to correct underestimation with a convergence guarantee, ensuring minimal constraint violations while maintaining performance.

**Overestimation in RL.** Overestimation in RL has been extensively studied. Double Q-learning (Hasselt, 2010) and Double DQN (Van Hasselt et al., 2016) reduce bias by using separate target value functions, avoiding maximization-induced errors. However, in actor-critic frameworks, the slow-changing policy keeps current and target values close, failing to fully eliminate maximization bias. TD3 (Fujimoto et al., 2018) addresses this by using a pair of critics and selecting the minimum value. AdaEQ (Wang et al., 2021) employs an ensemble-based method, adjusting the ensemble size based on Q-value approximation error to mitigate overestimation. Notably, overestimation and underestimation biases can be beneficial in certain scenarios. Methods like Maxmin Q-learning (Lan et al., 2020) and Balanced Q-learning (Karimpanal et al., 2023) focus on controlling estimation bias to enhance learning. In this paper, we demonstrate that underestimation of the cost value causes constraint violations during training and introduce the intrinsic cost to mitigate underestimation, controlling bias for safer exploration.

**Intrinsic reward.** Intrinsic rewards in RL are typically used to enhance exploration, falling into two categories: encouraging exploration of novel states (Zhang et al., 2021; Seo et al., 2021), and reducing prediction errors or uncertainties to improve environmental understanding (Sharma et al., 2019; Laskin et al., 2022). Count-based methods (Strehl & Littman, 2008; Badia et al., 2020) use state visit counts as bonus rewards to encourage the exploration of new states. (Lipton et al., 2016) indicates that agents tend to periodically revisit states under new policies after forgetting them, introducing an intrinsic fear model to prevent periodic catastrophes. In CRL, ROSARL (Tasse et al., 2023) interprets constraints as intrinsic rewards, optimizing policies by assigning minimal penalties to unsafe states. This paper introduces intrinsic costs to enhance safer exploration, particularly when extrinsic costs are underestimated. Memory-driven intrinsic costs provide anticipatory signals, guiding policy updates to avoid revisiting high-cost regions.

## 3. Preliminary

CRL can be modeled as a Constrained Markov decision process (CMDP), denoted by a tuple $(S, A, R, C, P, \rho, \gamma)$, where $S$ is the state space, $A$ is the action space, $R : S \times A \to \mathbb{R}$ is the reward function, $P : S \times A \to [0, 1]$ is the transition probability function, $\rho$ is the initial state distri-

bution, and $\gamma \in (0, 1)$ is the discount factor. The extrinsic cost function $C : S \times A \to \mathbb{R}$ maps state-action pairs to extrinsic costs $c^E$, which are task-specific costs provided by the environment. The intrinsic cost is denoted as $c^I$, and $d$ denotes the constraints threshold.

Starting from an initial state $s_0$ sampled from the initial state distribution $\rho$, the agent perceives the state $s_t$ from the environment at each time step $t$, selects an action $a_t$ according to the policy $\pi : S \to A$, receives the reward $r_t = R(s_t, a_t)$ and extrinsic cost $c_t^E$, and transfers to the next state $s_{t+1}$ based on $P(s_{t+1}|s_t, a_t)$. The set of all stationary policies is denoted as $\Pi$. The discounted future state visitation distribution is defined as $d^\pi(s) := (1 - \gamma) \sum_{t=0}^{\infty} \gamma^t P(s_t = s|\pi)$. The value function for a policy $\pi$ is $V_R^\pi(s) := \mathbb{E}_{\tau \sim \pi}[\sum_{t=0}^{\infty} \gamma^t R(s_t, a_t)|s_0 = s]$, and action-value function is $Q_R^\pi(s, a) := \mathbb{E}_{\tau \sim \pi}[\sum_{t=0}^{\infty} \gamma^t R(s_t, a_t)|s_0 = s, a_0 = a]$. The advantage function measures the advantage of action $a$ over the mean value: $A_R^\pi(s, a) := Q_R^\pi(s, a) - V_R^\pi(s)$. The cost value function $V_C^\pi(s)$, cost action value function $Q_C^\pi(s, a)$ and cost advantage function $A_C^\pi(s, a)$ in CMDP can be obtained as in MDP by replacing the reward $r$ with the cost $c$. The expected discounted return $J_R(\pi) := \mathbb{E}_{\tau \sim \pi}[\sum_{t=0}^{\infty} \gamma^t R(s_t, a_t)]$, and the expected cumulative discounted cost $J_C(\pi) := \mathbb{E}_{\tau \sim \pi}[\sum_{t=0}^{\infty} \gamma^t C(s_t, a_t)]$, where $\tau = (s_0, a_0, s_1, a_1, \cdots)$ is the trajectory under $\pi$.

The CRL aims to find an optimal policy by maximizing the expected discounted return over the set of feasible policies $\Pi_C := \{\pi \in \Pi : J_C(\pi) \leq d\}$:

$$\arg \max_{\pi \in \Pi} J_R(\pi)$$
$$s.t. \quad J_C(\pi) \leq d \tag{1}$$

## 4. Methodology

In this section, we introduce the MICE algorithm. We first present the underestimation in the CRL cost value function. Then we introduce the flashbulb memory mechanism for recording high-cost regions, and derive the intrinsic cost to correct underestimation. Finally, we propose an extrinsic-intrinsic update formulation for MICE and a new optimization objective with its solutions.

### 4.1. Underestimate Bias in Cost Value Function

Overestimation in RL arises from the tendency of the value function update to greedily select high value actions, resulting in estimates exceeding the optimal value (Fujimoto et al., 2018). Conversely, the cost value function in CRL exhibits underestimation, particularly during constraint violations, due to the inherent tendency to minimize costs.

In methods like Q-learning, the cost value function is updated using a greedy strategy during constraint violations: $Q_C(s, a) \leftarrow Q_C(s, a) + \alpha[c + \gamma \min_{a'} Q_C(s', a') -$

$Q_C(s, a)]$, where $\alpha$ is the step size. Assuming that value estimates contain zero-mean noise $\epsilon$, a consistent underestimation bias is induced by minimizing the noisy value estimate $Q_C(s', a') + \epsilon$. The zero-mean property of noise is disrupted after minimization, resulting in the minimized value estimate being generally smaller than the true minimum (Thrun & Schwartz, 2014):

$$\mathbb{E}_\epsilon[\min_{a'} Q_C(s', a') + \epsilon] \leq \min_{a'} Q_C(s', a') \tag{2}$$

Such noise is inherent in function approximation methods (Fujimoto et al., 2018).

In actor-critic-based CRL methods, the policy learns from value estimation provided by the approximate reward critic and cost critic. When constraints are violated, the policy is updated with a policy gradient that minimizes the expected cost value estimate: $\arg \min_{\pi \in \Pi_\theta} \mathbb{E}_{s \sim d^\pi, a \sim \pi}[Q_C(s, a)]$, where $\Pi_\theta$ represents the policy set parameterized by $\theta$. Denote the true cost value function as $Q_C^*(s, a)$ and the approximate cost value function as $\hat{Q}_C(s, a)$. Updated from the current policy $\pi_k(\cdot|\theta)$ with deterministic policy gradient, let $\pi$ be the policy derived from the true cost value $Q_C^*(s, a)$, and $\hat{\pi}$ the policy derived from the approximate cost value $\hat{Q}_C(s, a)$. According to TD3 (Fujimoto et al., 2018), when the approximation introduces a bias such that $\mathbb{E}[\hat{Q}_C(s, \pi(s))] \leq \mathbb{E}[Q_C^*(s, \pi(s))]$ due to unavoidable noise in the function approximation, then the cost value is underestimated under the updated policy $\hat{\pi}$ within a sufficiently small step size:

$$\mathbb{E}[\hat{Q}_C(s, \hat{\pi}(s))] \leq \mathbb{E}[Q_C^*(s, \hat{\pi}(s))] \tag{3}$$

To validate the presence of underestimation bias, we compare the estimated cost value for various states against their corresponding true values in both the primal method CPO (Achiam et al., 2017) and the primal-dual method PID Lagrangian (Stooke et al., 2020). The true values are computed by averaging the cumulative discounted costs over $1,000$ episodes under the current policy. Results in Figure 2 show that cost value functions in different CRL methods are consistently and significantly underestimated across various environments during training.

Compared to overestimation in RL, underestimation in CRL can have more detrimental impacts, as it directly leads to unsafe actions that violate constraints. When the cost critic underestimates the true cost, actions with high rewards but violating constraints are mistakenly perceived as safe, causing these unsafe actions to be selected. Then these actions are propagated through the Bellman equation, accumulating bias and generating increasingly unsafe policies. This explains the frequent constraint violations observed during training in various CRL methods.

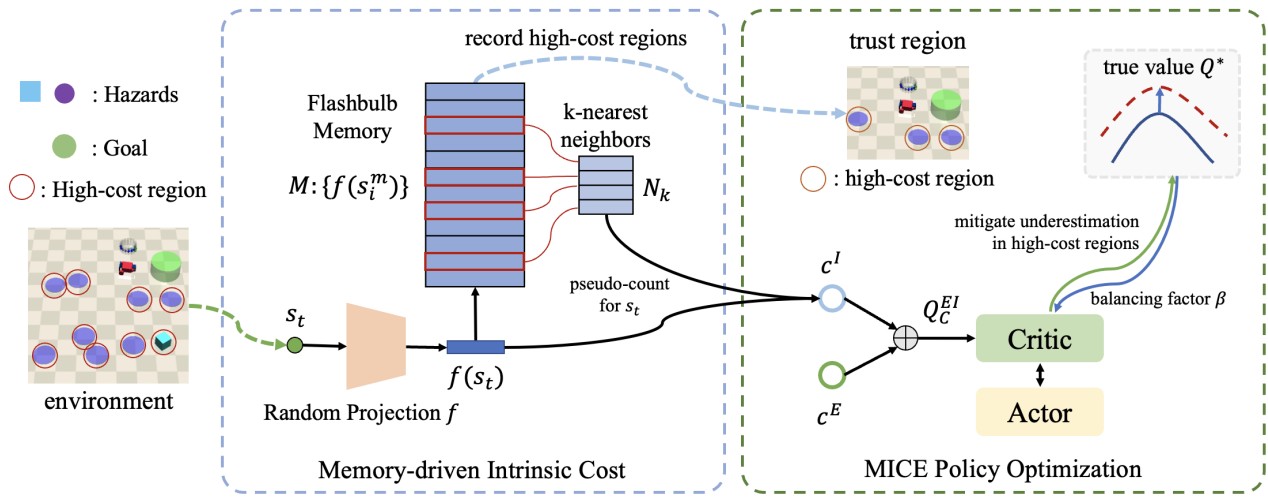

Figure 1: Structure of MICE. Underestimation of cost value in high-cost regions causes constraint violations. The Flashbulb Memory records high-cost regions by storing previously explored unsafe states. The intrinsic cost is computed through the pseudo-count of the current state's visit to high-cost regions in memory. The trust region ensures the alignment of the stored high-cost regions with the current policy. The extrinsic-intrinsic cost critic mitigates underestimation in high-cost regions.

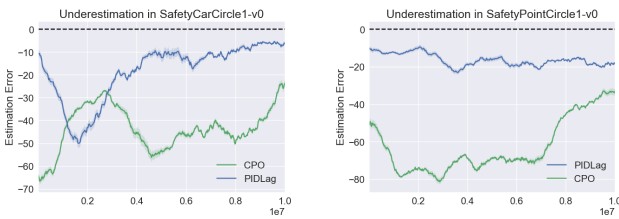

Figure 2: Underestimation bias across environments. The x-axis is time steps, the y-axis is the cost value estimate minus the true value, and the dashed line is the zero deviation.

### 4.2. Intrinsic Cost Generated from Flashbulb Memory

Risk awareness enables humans to identify potential dangers and adopt conservative behaviors to ensure safety. Humans are impressed by previous risky experiences, which are vividly recalled to avoid danger in similar scenarios. However, CRL agents with underestimated critics often fail to recognize the consequences of unsafe actions, leading to constraint violations. Inspired by human cognitive mechanisms, we introduce a memory-driven intrinsic cost mechanism to enhance the agent's risk awareness. This intrinsic cost discourages repeated visits to unsafe states while promoting safer exploration.

We construct a flashbulb memory $M$ to record high-cost regions by storing unsafe states $s^m$ with positive extrinsic costs. To reduce dimensionality while approximately preserving inter-sample distances, each state $s$ is embedded into a lower-dimensional latent space $f(s)$ using a random projection layer $f$, implemented with a Gaussian random

matrix (Zhu et al., 2020). By the Johnson–Lindenstrauss lemma (Johnson et al., 1984), this mapping approximately maintains relative Euclidean distances in the original space. The memory, with dynamically adjusted capacity, stores unsafe samples collected by the policy from the previous iteration. By mirroring the human tendency to prioritize recent high-risk experiences, this design ensures that the memory remains aligned with the current policy.

The intrinsic cost $c^I$ is derived from the flashbulb memory. By comparing the current state $s_t$ with states stored in memory, smaller differences indicate higher similarity to prior unsafe experiences, signaling an increased risk of constraint violations and resulting in a higher intrinsic cost $c^I$. Denote the flashbulb memory as $M$ : $\{f(s_0^m), \cdots, f(s_i^m), \cdots\}$, where $s_i^m$ is the $i$-th unsafe state. Inspired by theoretically-justified exploration methods that convert state-action counts into bonus rewards (Strehl & Littman, 2008), the intrinsic cost $c_t^I$ at time $t$ is generated by comparing the current embedding state $f(s_t)$ with those in memory $M$:

$$c_t^I = \sqrt{\sum_{s_i^m \sim N_k} K(f(s_t), f(s_i^m))} \qquad (4)$$

Drawing inspiration from NGU (Badia et al., 2020), the sum of the similarities, computed using a kernel function $K$, is used to represent the pseudo-count of the current state $s_t$ visiting high-cost regions in memory. This sum is approximated by the $k$-nearest neighbors $N_k$ of $f(s_t)$ in $M$. The kernel function is:

$$K(x, y) = \frac{\xi}{l^2(x, y) + \xi} \qquad (5)$$

where $l$ is the Euclidean distance, and $\xi$ is a small constant for numerical stability (set to $10^{-3}$ following NGU). To preserve training stability, the intrinsic cost is normalized. A larger visit count for $s_t$ indicates a higher probability that it resides in a high-cost region, thus resulting in a greater intrinsic cost.

### 4.3. Safety Policy Optimization with the Intrinsic Cost

The standard cost value function $Q_C(s, a)$ in CRL is updated using only extrinsic costs $c^E$: $Q_C(s, a) = (1 - \alpha)Q_C(s, a) + \alpha(c^E + \gamma\mathbb{E}_{s'}[Q_C(s', a')])$. To mitigate underestimation, we propose an extrinsic-intrinsic cost value function $Q_C^{EI}(s, a)$, which incorporates memory-driven intrinsic costs $c^I$ and task-driven extrinsic costs $c^E$:

$$Q_C^{EI}(s, a) = (1 - \alpha)Q_C^{EI}(s, a) \\ + \alpha(c^E + \beta c^I + \gamma\mathbb{E}_{s'}[Q_C^{EI}(s', a')]) \quad (6)$$

where $\alpha$ is the step size, $\beta$ is the intrinsic factor. Starting from the same initialization, $Q_C^{EI}(s, a) \geq Q_C(s, a)$ for the same state-action pair, as the target $Q_T = c^E + \beta c^I + \gamma\mathbb{E}_{s'}[Q_C^{EI}(s', a')]$ for $Q_C^{EI}$ incorporates the intrinsic cost, resulting in a larger target compared to the standard cost estimate. By augmenting the agent's memory, the extrinsic-intrinsic target cost value increases the cost estimate, effectively mitigating underestimation in high-cost regions.

Overestimation and underestimation biases in cost value are not inherently detrimental, their impact depends on the current state (Lan et al., 2020), with detailed analysis in Appendix B.1. Generally, underestimation bias encourages exploration, while overestimation bias tends to discourage exploration. In high-cost regions with a high risk of constraint violation, overestimation bias can be beneficial by discouraging unsafe exploration, while underestimation bias may lead to over-exploration of these unsafe regions.

In this work, flashbulb memory is employed to identify previously explored high-cost regions, with intrinsic cost quantifying the pseudo-count of the current state visiting these regions. Higher intrinsic costs signal a greater likelihood of entering a high-cost region, potentially introducing overestimation to discourage unsafe explorations. Additionally, the propagation of overestimation through cost value updates is limited, as the policy tends to avoid actions with large cost estimates (Fujimoto et al., 2018).

To avoid excessive estimate bias, we propose an adaptive bias correction mechanism. The target for the $(n + 1)$-th update $Q_{T_{n+1}}$ is modified as:

$$Q'_{T_{n+1}} = Q_{T_{n+1}} - \alpha(Q_n - Q^*) \quad (7)$$

where $Q^*$ is the optimal value, $Q_n$ is the estimate at the $n$-th update. When $Q_n - Q^* < 0$, the modified target $Q'_{T_{n+1}}$ is adjusted upward, introducing positive bias to mitigate

excessive underestimation. Conversely, when $Q_n - Q^* > 0$, $Q'_{T_{n+1}}$ is adjusted downward, introducing negative bias to address excessive overestimation. Then we derive the balancing intrinsic factor $\beta$ to control the estimation bias.

**Proposition 4.1.** *For a transition $(s, a, c^E, s')$ in a CMDP, where the $n$-th update of the extrinsic-intrinsic cost value $Q_n$ corresponds to the $n$-th update target $Q_{T_n}$, the modified target for the $(n + 1)$-th update is $Q'_{T_{n+1}}$, the balancing intrinsic factor $\beta'$ for the $(n + 1)$-th update is given by:*

$$\beta' = max\{\gamma^n(\beta_n - \frac{\alpha\epsilon_n}{c^I}), 0\}, \quad c^I > 0 \quad (8)$$

*where $\epsilon_n = Q_n - Q^*$ is the $n$-th update estimation bias, $\gamma \in (0, 1)$ is the discount factor.*

The derivation process is provided in Appendix B.2.

The direction of the $\beta$ update depends solely on $\epsilon_n$. When $\epsilon_n < 0$, indicating underestimation of $Q_n$, $\beta'$ is increased to raise the intrinsic cost and mitigate underestimation. When $\epsilon_n > 0$, indicating overestimation of $Q_n$, $\beta'$ is decreased, reducing the intrinsic cost to mitigate overestimation. The discount factor is used to ensure the convergence of the value function, with the convergence analysis in Theorem 4.5. By leveraging the estimated bias from the previous update, we iteratively refine the intrinsic factor to control the estimated bias of the current value function.

In practical implementation, since the optimal value $Q^*$ is unknown, we approximate it by computing the cumulative discounted cost along trajectories sampled by the current policy, following (Wang et al., 2021; Fujimoto et al., 2018).

Based on the extrinsic-intrinsic cost value function, the cumulative discounted extrinsic-intrinsic cost is defined as:

$$J_C^{EI}(\pi) := \mathbb{E}_{\tau\sim\pi}\left[\sum_{t=0}^{\infty}\gamma^t C^{EI}(s_t, a_t)\right] \quad (9)$$

where $C^{EI}(s_t, a_t) = c_t^E + \beta c_t^I$ is the extrinsic-intrinsic cost function. The extrinsic-intrinsic advantage function in MICE is defined as: $A_C^{EI}(s, a) = \mathbb{E}_{s'}[c^E + \beta c^I + \gamma V_C(s') - V_C(s)]$. To minimize constraint violations, we replace the standard constraint $J_C$ with the extrinsic-intrinsic constraint $J_C^{EI}$ in the optimization objective. To facilitate optimization, we give the difference in expectation constraint between extrinsic-intrinsic $J_C^{EI}(\pi')$ and extrinsic $J_C(\pi)$.

**Lemma 4.2.** *Given arbitrary two policies $\pi$ and $\pi'$, the difference in expectation constraint of extrinsic-intrinsic $J_C^{EI}(\pi')$ and extrinsic $J_C(\pi)$ is expressed as:*

$$J_C^{EI}(\pi') - J_C(\pi) = \mathbb{E}_{\tau|\pi'}\left[\sum_{t=0}^{\infty}\gamma^t A_C^{EI}(s_t, a_t|\pi)\right] \quad (10)$$

*where $A_C^{EI}(s_t, a_t|\pi) = \mathbb{E}_{s_{t+1}}[c_t^E + \beta c_t^I + \gamma V_C^{\pi}(s_{t+1}) - V_C^{\pi}(s_t)]$. The expectation is over trajectories $\tau$, with $\mathbb{E}_{\tau|\pi'}$ indicating that actions are sampled from $\pi'$ to generate $\tau$.*

A proof is provided in Appendix B.3.

According to Equation 10, we present the optimization objective of MICE:

$$\pi_{k+1} = \arg \max_{\pi \in \Pi_\theta} \mathbb{E}_{s \sim d^\pi, a \sim \pi}[A_R^{\pi_k}(s, a)]$$

$$s.t. \quad J_C(\pi_k) + \frac{1}{1 - \gamma} \mathbb{E}_{s \sim d^\pi, a \sim \pi}[A_C^{EI}(s, a|\pi_k)] \leq d \tag{11}$$

To ensure that the unsafe states stored in memory remain relevant to the updated policy $\pi$, we further propose a surrogate objective within the trust region:

$$\pi_{k+1} = \arg \max_{\pi \in \Pi_\theta} \mathbb{E}_{s \sim d^{\pi_k}, a \sim \pi}[A_R^{\pi_k}(s, a)]$$

$$s.t. \quad J_C(\pi_k) + \frac{1}{1 - \gamma} \mathbb{E}_{s \sim d^{\pi_k}, a \sim \pi}[A_C^{EI}(s, a|\pi_k)] \leq d,$$

$$D(\pi \| \pi_k) \leq \varphi \tag{12}$$

where $D(\pi \| \pi_k) = \mathbb{E}_{s \sim d^{\pi_k}}[D_{KL}(\pi \| \pi_k)[s]]$, and $D_{KL}$ is the KL divergence. $\varphi > 0$ is the trust region size, and the set $\{\pi \in \Pi_\theta : D(\pi \| \pi_k) \leq \varphi\}$ defines the trust region.

The trust region constrains the updated policy $\pi$ to remain close to the previously sampled policy $\pi_k$. This alignment preserves the relevance of unsafe states stored in memory, ensuring that the identified high-cost regions adequately cover the sampling space of the updated policy $\pi$. Consequently, the intrinsic cost more accurately reflects the risk of the updated policy $\pi$ visiting these high-cost regions, effectively mitigating underestimation and promoting safer exploration in unsafe regions.

To solve the optimization objective 12, we propose the MICE-CPO and MICE-PIDLag optimization methods, as detailed in Appendix A.

### 4.4. Theoretical Analysis

Theoretically, we establish an upper bound on the difference in the expected extrinsic–intrinsic constraint between two arbitrary policies.

**Theorem 4.3** (Extrinsic-intrinsic Constraint Bounds). *For arbitrary two policies $\pi'$ and $\pi$, the following bound for the expected extrinsic-intrinsic constraint holds:*

$$J_C^{EI}(\pi') - J_C^{EI}(\pi)$$

$$\leq \frac{1}{1 - \gamma} \mathbb{E}_{s \sim d^\pi, a \sim \pi'} \left[ A_C^{EI}(s, a|\pi) + \frac{2\gamma \epsilon_{\pi'}^{EI}}{1 - \gamma} D_{TV}(\pi' \| \pi)[s] \right] \tag{13}$$

*where $\epsilon_{\pi'}^{EI} := \max_s |\mathbb{E}_{a \sim \pi'}[A_C^{EI}(s, a|\pi)]|$, the TV-divergence $D_{TV}(\pi' \| \pi)[s] = (1/2) \sum_a |\pi'(a|s) - \pi(a|s)|$.*

The proof is provided in Appendix B.4.

The upper bound in Theorem 4.3 is associated with the TV divergence between $\pi$ and $\pi'$. A larger divergence between two policies results in a larger upper bound on the constraint gap. This theorem supports the optimization objective 12 within the trust region in MICE.

By mitigating the underestimation, the MICE algorithm significantly reduces constraint violations during the learning process. We further establish a theoretical upper bound on the constraint violation for the updated policy within the MICE optimization framework:

**Theorem 4.4** (MICE Update Worst-Case Constraint Violation). *Suppose $\pi_k$, $\pi_{k+1}$ are related by the optimization objective 12, an upper bound on the constraint of the updated policy $\pi_{k+1}$ is:*

$$J_C(\pi_{k+1}) \leq d - I + \frac{\sqrt{2\varphi}\gamma \epsilon_C^{\pi_{k+1}}}{(1 - \gamma)^2} \tag{14}$$

*where $\epsilon_C^{\pi_{k+1}} := \max_s |\mathbb{E}_{a \sim \pi_{k+1}}[A_C^{\pi_k}(s, a)]|$, and $I = \mathbb{E}_{\tau|\pi_{k+1}} \left[ \sum_{t=0}^\infty \gamma^t \beta c_t^I \right]$.*

A proof is provided in Appendix B.4.

Theorem 4.4 demonstrates that our method achieves a tighter upper bound on constraint violation compared to CPO, guaranteeing that the updated policy in MICE has a lower probability of exceeding the constraint threshold.

Based on similar assumptions as in TD3 and Double Q-learning, we give convergence guarantees of the extrinsic-intrinsic cost value function.

**Theorem 4.5** (Convergence Analysis). *Given following conditions:*

1. *Each state-action pair is sampled an infinite number of times.*

2. *The MDP is finite.*

3. *$\gamma \in [0, 1)$.*

4. *$Q_C^{EI}$ values are stored in a lookup table.*

5. *$Q_C^{EI}$ receives an infinite number of updates.*

6. *The learning rates satisfy $\alpha_t(s, a) \in [0, 1]$, $\sum_t \alpha_t(s, a) = \infty$, $\sum_t (\alpha_t(s, a))^2 < \infty$ with probability 1, $\alpha_t(s, a) = 0, \forall (s, a) \neq (s_t, a_t)$.*

7. *$Var[c_t^E + \beta c_t^I] < \infty, \forall s, a$.*

*The extrinsic-intrinsic $Q_C^{EI}$ will converge to the optimal $Q_C^*$ with probability 1.*

The proof is in Appendix B.5.

Theorem 4.5 ensures that MICE converges to the optimal solution, which guarantees policy performance while satisfying constraints.

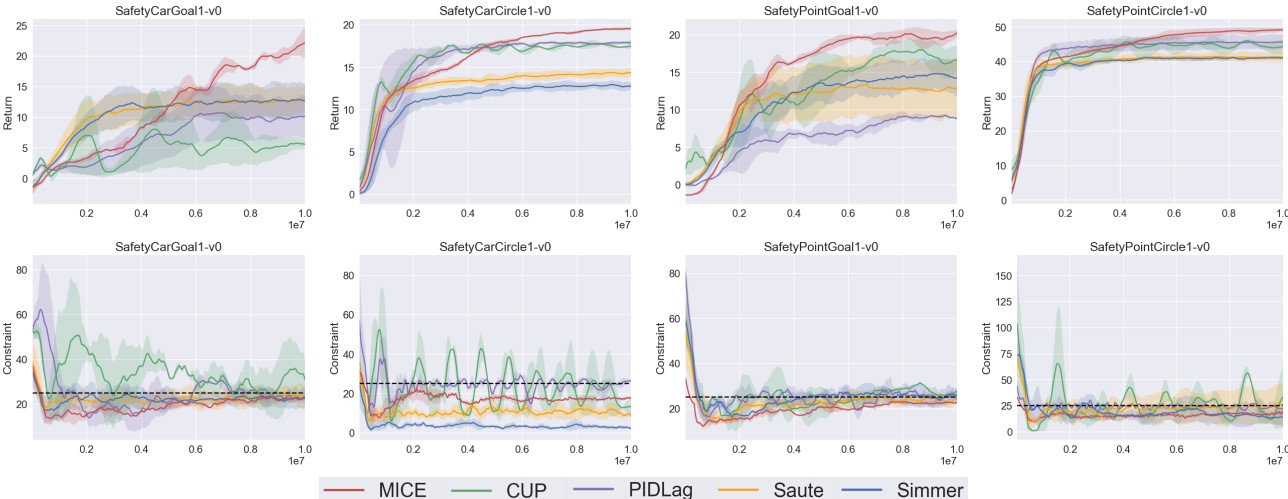

Figure 3: Comparison of MICE to baselines on Safety Gym. The x-axis is the total number of training steps, the y-axis is the average return or constraint. The solid line is the mean and the shaded area is the standard deviation. The dashed line is the constraint threshold which is 25.

## 5. Experiment

The experiments address the following questions: 1) Does MICE reduce constraint violations while maintaining policy performance compared to baselines? 2) Does the intrinsic cost component effectively mitigate underestimation? Baselines include the primal-dual PID Lagrangian method (Stooke et al., 2020), the primal CUP method (Yang et al., 2022), and state augmentation methods Saute (Sootla et al., 2022a) and Simmer (Sootla et al., 2022b) focusing on zero constraint violations. Results for additional baselines are provided in the Appendix C.3.1.

We implemented both MICE-CPO and MICE-PIDLag, with optimization details provided in Appendix A. All experiments followed uniform conditions to ensure fairness and reproducibility, with a total of $10^7$ training time steps and a maximum trajectory length of $1000$ steps. To reduce randomness, 6 random seeds were used for each method, and the results are presented as mean and variance. The algorithm process is provided in Appendix C.1, and additional experiments are presented in Appendix C.3.

**Environments Description.** Experiments were conducted across four navigation tasks in Safety Gym (Ji et al., 2023) and four MuJoCo physical simulator tasks (Todorov et al., 2012), as detailed in Appendix C.4. All tasks aim to maximize expected reward (higher is better) while satisfying constraints (below a threshold). In Safety Gym, we train Point and Car agents on navigation tasks, including the Goal task (navigate to a goal while avoiding hazards) and the Circle task (go around a circle's center without crossing boundaries). In Safety MuJoCo, agents are rewarded for running along a straight path with a velocity limit for safety.

**Performance and Constraint.** Figure 3 presents the learning curves for MICE and baselines in Safety Gym. The first row shows the cumulative discounted reward during training, and the second row is the cumulative discounted cost, with the black dashed line indicating the cost threshold. The results indicate that MICE significantly reduces constraint violations while maintaining superior or similar policy performance to baselines. Notably, in the Goal1 navigation tasks with multiple hazards, MICE enhances policy performance while maintaining constraint satisfaction, as its intrinsic cost provides predictive signals to avoid hazards and encourage safe exploration. In Safety MuJoCo, as shown in Figure 4, MICE consistently maintains zero constraint violations throughout training, with convergence speed comparable to or faster than baselines. MICE matches the constraint satisfaction levels of Saute and SimmerPID, which emphasize zero constraint violations, while surpassing them in policy performance. In HalfCheetahVelocity, PIDLag exceeds the constraint threshold, thus its higher return compared to MICE does not indicate a better policy. Extended experiments covering more complex tasks are provided in Appendix C.3.2. The results demonstrate that MICE continues to achieve superior constraint satisfaction while maintaining policy performance in these more challenging scenarios.

**Estimation Bias in MICE.** To validate the effectiveness of the intrinsic cost in MICE for mitigating estimation bias, we compare the difference between estimated and true cost values across MICE and baselines. True values are computed as the average cumulative discounted cost over $1,000$ episodes under the current policy. Figure 5 shows that the estimated values in MICE are significantly higher than those of the baselines, indicating that the intrinsic cost effectively

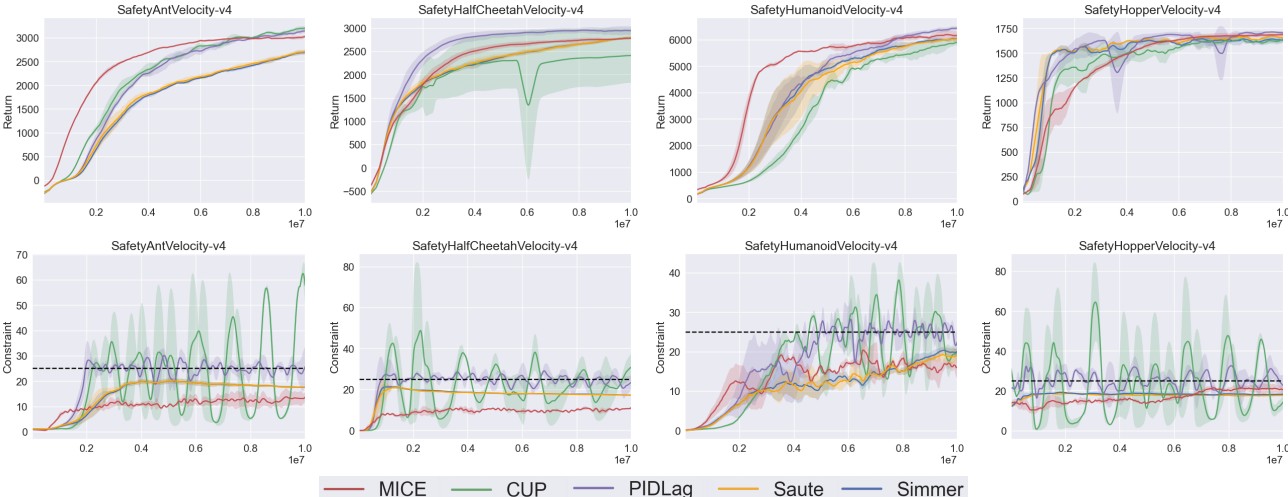

Figure 4: Comparison of MICE to baselines on Safety MuJoCo. The x-axis is the total number of training steps, the y-axis is the average return or constraint. The solid line is the mean and the shaded area is the standard deviation. The dashed line in the cost plot is the constraint threshold which is 25.

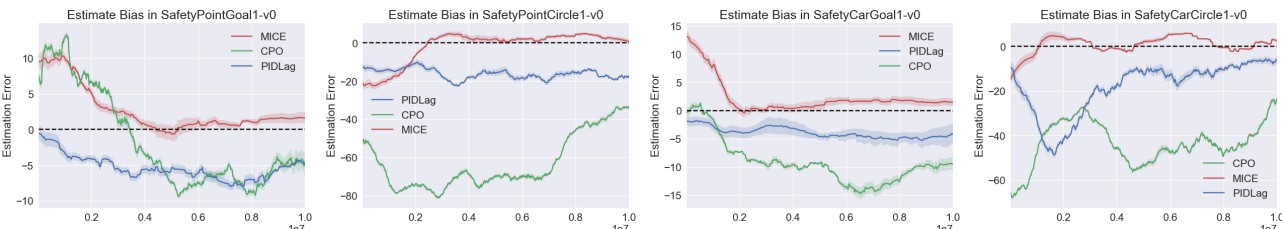

Figure 5: Validation experiments of mitigating underestimation with MICE. The y-axis is the cost value estimate minus the true value, and the dashed line is the zero deviation.

mitigates underestimation. Meanwhile, the estimation bias in MICE approaches zero, confirming that the balancing factor $\beta$ in MICE can effectively correct estimation bias. Furthermore, the extrinsic-intrinsic cost value function in MICE gradually converges to the true value, supporting the convergence analysis in Theorem 4.5. Additionally, we compare the TD3-based cost value function with that of MICE in mitigating constraint bias, showing that the intrinsic cost component in MICE more accurately mitigates underestimation, resulting in improved constraint satisfaction and policy performance, as detailed in Appendix C.3.3.

**Ablation Study of Intrinsic Cost.** To verify that memory-driven intrinsic cost enhances policy learning, we conduct an ablation study where the intrinsic cost in MICE is replaced by fixed constants (3, 5, 15). Figure 6a and 6b show that introducing fixed constants results in decreased policy performance. Notably, constraint violations are more frequent with constant 5 compared to 3, indicating that simply adding a larger value to the cost estimate does not necessarily reduce constraint violations. MICE achieves the best performance, demonstrating that the intrinsic cost in MICE does not merely ensure constraint satisfaction by

introducing conservatism. Instead, it is derived from the count of visits to high-regions stored in memory, effectively mitigating underestimation in high-regions and promoting safer exploration. Our method is theoretically and empirically proven to converge to the optimal value, while the constant-based approach lacks such guarantees, leading to suboptimal policy performance. Additionally, we perform an ablation study on the random projection layer in MICE to validate its effectiveness. The results show that it significantly reduces computational complexity without degrading policy performance or increasing constraint violations, as detailed in Appendix C.3.4.

**Robustness to Constraint Thresholds.** Sensitivity analysis experiments are conducted in SafetyPointGoal1-v0 with thresholds of 0, 15, and 25, as illustrated in Figure 6c and 6d. The results show that MICE adapts effectively to varying constraint requirements. With a threshold of 15, MICE achieves a balance between performance and constraint satisfaction. Under the strict threshold of 0, MICE enforces compliance, ensuring the policy adheres to constraints.

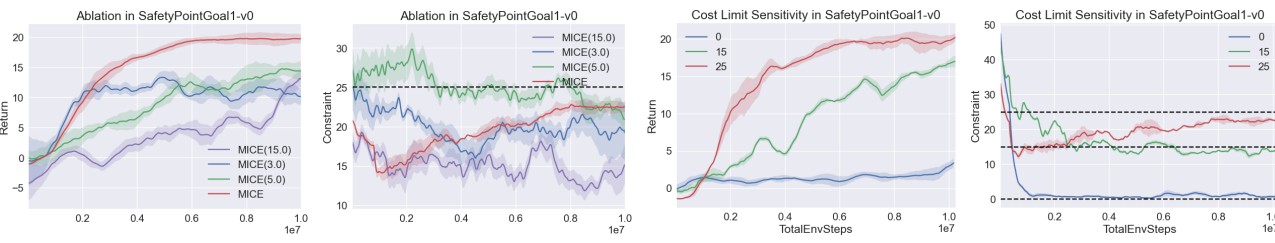

(a) Return of Ablation      (b) Constraint of Ablation      (c) Return of Sensitivity      (d) Constraint of Sensitivity

Figure 6: **(a)(b)** Ablation study of extrinsic-intrinsic cost value in MICE, comparing MICE with variants that remove the intrinsic cost and add fixed constants to the cost value function. **(c)(d)** Robustness of MICE to different cost thresholds.

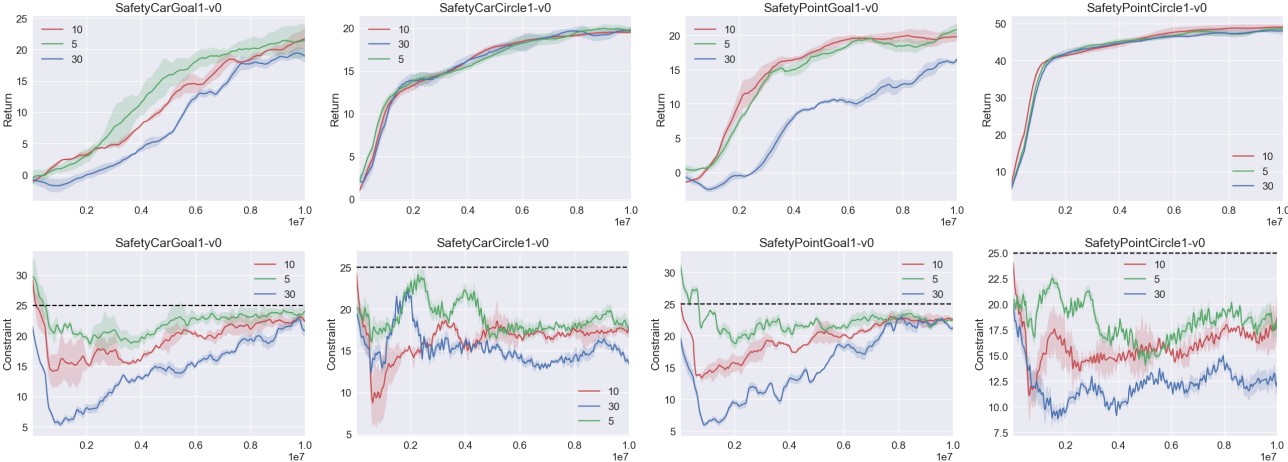

Figure 7: Sensitivity analysis of MICE algorithm for the number of nearest neighbors $k$ in KNN, showing that increasing $k$ enhances the safety of the policy but also raises computational overhead. We uniformly set $k = 10$ across all environments to balance safety, performance, and computational efficiency.

**Sensitivity Analysis of Hyperparameters.** To evaluate the sensitivity of the number of nearest neighbors $k$ in KNN, we varied $k$ across multiple environments and analyzed its impact, as shown in Figure 7. The results indicate that increasing $k$ enhances policy safety by incorporating more unsafe states from memory, but also raises computational overhead. Conversely, a smaller $k$ may result in insufficient leveraging of unsafe state information, potentially leading to higher constraint violations. In this paper, we uniformly set $k = 10$ across all environments to balance safety, performance, and computational efficiency.

## 6. Discussion and Conclusion

This paper addresses an important challenge in CRL, the underestimation of the cost value, which significantly contributes to constraint violations. To mitigate this issue, we propose the MICE algorithm, which incorporates an extrinsic-intrinsic cost value update mechanism inspired by human cognitive processes. MICE enhances the cost estimates of high-cost regions, encouraging safer exploration. Theoretically, we provide an upper bound on constraint violations and establish convergence guarantees of the MICE

algorithm. Extensive experimental results demonstrate that MICE effectively reduces constraint violations while maintaining robust policy performance.

The MICE framework exhibits strong applicability across diverse tasks. For example, the criterion for adding states to the flashbulb memory can be flexibly adjusted. In our experiments, we adopt a rule based on whether the extrinsic cost of a state is greater than zero, where extrinsic costs are directly provided by the environment without noise. However, in real-world applications, extrinsic costs may be noisy and may not accurately reflect the true cost of a given state. A practical alternative is to employ the expected cumulative cost as the criterion for memory inclusion: a state is stored if its expected cumulative cost exceeds a predefined threshold. By aggregating information across multiple future timesteps, this approach effectively mitigates the influence of individual outlier values. Additionally, in domains such as autonomous driving, the expense of sampling unsafe states is prohibitively high. In these scenarios, the memory can be constructed using offline datasets to ensure safety and feasibility, with techniques such as importance resampling employed to correct for potential distributional shifts.

## Acknowledgment

This work was supported by NSF China under Grant No. T2421002, 62202299, 62020106005, 62061146002.

## Impact Statement

This paper presents work whose goal is to advance the field of Reinforcement Learning. By introducing the MICE algorithm, we substantially enhance safety during RL training. In real-world applications, MICE can facilitate safer and more efficient deployment of RL in domains such as large language models, autonomous driving, and robotics, ultimately advancing the field of Machine Learning. There are potential societal consequences of our work, none of which we feel must be specifically highlighted here.

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

## Notations

| | |
|---|---|
| $c^I$ | intrinsic cost |
| $c^E$ | extrinsic cost |
| $R$ | reward function |
| $P$ | transition probability function |
| $\rho$ | initial state distribution |
| $\gamma$ | discount factor |
| $\pi$ | the policy |
| $d^\pi$ | the discounted future state visitation distribution |
| $\tau$ | trajectory |
| $A_R^\pi(s,a)$ | advantage function |
| $J_R$ | the expected discount return |
| $J_C$ | the expected discount cost return |
| $d$ | cost threshold |
| $\Pi_C$ | the set of feasible policies |
| $Q_C^\pi(s,a)$ | action cost value function |
| $V_C^\pi(s)$ | cost value function |
| $A_C^\pi(s,a)$ | cost advantage function |
| $\alpha$ | step size |
| $M$ | flashbulb memory |
| $s^m$ | unsafe state |
| $f$ | Random Projection |
| $\beta$ | intrinsic factor |
| $\epsilon$ | estimation bias |
| $Q_C^{EI}$ | extrinsic-intrinsic cost value function |
| $Q_T$ | extrinsic-intrinsic update target |
| $Q_n$ | extrinsic-intrinsic $n$-th update |
| $J_C^{EI}$ | cumulative discount extrinsic-intrinsic cost |
| $C^{EI}$ | extrinsic-intrinsic cost function |
| $A_C^{EI}$ | extrinsic-intrinsic cost advantage function |
| $\delta$ | extrinsic-intrinsic TD-error |
| $\varphi$ | trust region size |
| $k$ | the number of nearest neighbors in KNN |

## A. Safety Policy Optimization in MICE

The CRL aims to find an optimal policy by maximizing the expected discount return over the set of feasible policies $\Pi_C := \{\pi \in \Pi : J_C(\pi) \leq d\}$:

$$\arg \max_{\pi \in \Pi} J_R(\pi)$$
$$s.t. \quad J_C(\pi) \leq d \tag{15}$$

The following equation briefly gives the performance difference of arbitrary two policies, which represents the expected return of another policy $\pi'$ in terms of the advantage function over $\pi$:

$$J_R(\pi') - J_R(\pi) = \frac{1}{1-\gamma} \mathbb{E}_{s \sim d^{\pi'}, a \sim \pi'}[A_R^{\pi}(s, a)] \tag{16}$$

This implies that iterative updates to the policy, $\pi'(s) = \arg \max_a A_R^{\pi}(s, a)$, lead to performance improvement until convergence to the optimal solution.

According to the performance difference equation (16), CRL is defined as a constrained optimization problem:

$$\pi_{k+1} = \arg \max_{\pi \in \Pi_\theta} \mathbb{E}_{s \sim d^\pi, a \sim \pi}[A_R^{\pi_k}(s, a)]$$
$$s.t. \quad J_C(\pi_k) + \frac{1}{1-\gamma} \mathbb{E}_{s \sim d^\pi, a \sim \pi}[A_C^{\pi_k}(s, a)] \leq d \tag{17}$$

where policy $\pi \in \Pi_\theta$ is parameterized with parameters $\theta$, and $\pi_k$ represents the current policy.

In this paper, we define the cumulative discount extrinsic-intrinsic cost as:

$$J_C^{EI}(\pi) := \mathbb{E}_{\tau \sim \pi}[\sum_{t=0}^{\infty} \gamma^t C^{EI}(s_t, a_t)] \tag{18}$$

where $C^{EI}(s, a) = c^E + \beta c^I$ is the extrinsic-intrinsic cost function. The extrinsic-intrinsic advantage function in MICE is defined as:

$$A_C^{EI}(s, a) = \mathbb{E}_{s'}[c^E + \beta c^I + \gamma V_C(s') - V_C(s)] \tag{19}$$

To reduce constraint violations, we replace $J_C$ with the extrinsic-intrinsic constraint $J_C^{EI}$ in the optimization objective. We give the optimization objective of MICE based on the extrinsic-intrinsic cost value estimate and Lemma 1:

$$\pi_{k+1} = \arg \max_{\pi \in \Pi_\theta} \mathbb{E}_{s \sim d^\pi, a \sim \pi}[A_R^{\pi_k}(s, a)]$$
$$s.t. \quad J_C(\pi_k) + \frac{1}{1-\gamma} \mathbb{E}_{s \sim d^\pi, a \sim \pi}[A_C^{EI}(s, a | \pi_k)] \leq d \tag{20}$$

where policy $\pi \in \Pi_\theta$ is parameterized with parameters $\theta$, and $\pi_k$ represents the current policy. We propose two optimization methods, MICE-CPO and MICE-PIDLag, based on CPO and PID Lagrangian respectively, to solve the optimization objective 20.

### A.1. MICE-CPO

The complex dependency of state visitation distribution $d^\pi(s)$ on unknown policy $\pi$ makes objective 20 difficult to optimize directly. To address this, this paper uses samples generated by the current policy $\pi_k$ to approximate the original problem locally. We seek to solve the following optimization problem in the trust region:

$$\pi_{k+1} = \arg \max_{\pi \in \Pi_\theta} \mathbb{E}_{s \sim d^{\pi_k}, a \sim \pi}[A_R^{\pi_k}(s, a)]$$
$$s.t. \quad J_C(\pi_k) + \frac{1}{1-\gamma} \mathbb{E}_{s \sim d^{\pi_k}, a \sim \pi}[A_C^{EI}(s, a | \pi_k)] \leq d$$
$$D(\pi \| \pi_k) \leq \varphi \tag{21}$$

where $\Pi_\theta$ is the policy set parameterized by parameter $\theta$, $D(\pi\|\pi_k) = \mathbb{E}_{s\sim d^{\pi_k}}[D_{KL}(\pi\|\pi_k)[s]]$, $D_{KL}$ is the KL divergence and $\varphi > 0$ is the trust region size. The set $\{\pi \in \Pi_\theta : D(\pi\|\pi_k) \leq \varphi\}$ is the trust region.

In the MICE-CPO method, we approximate the reward objective and cost constraints with first-order expansion and approximate the KL-divergence constraint with second-order expansion. The local approximation to equation 21 is:

$$
\begin{aligned}
\theta_{k+1} =& \arg\max_\theta g^T(\theta - \theta_k) \\
s.t. \quad & c + (g_C^{EI})^T(\theta - \theta_k) \leq 0 \\
& \frac{1}{2}(\theta - \theta_k)^T H(\theta - \theta_k) \leq \varphi
\end{aligned}
\tag{22}
$$

where $g$ denotes the gradient of the reward objective in equation 21, $g_C^{EI}$ denotes the gradient of extrinsic-intrinsic constraint in equation 21, $c = J_C(\pi_k) - d$, $H$ is the Hessian of the KL-divergence. When the constraint is satisfied, we can get the analytical solution with the primal-dual method. The solution to the primal problem is:

$$
\theta^* = \theta_k + \frac{1}{\lambda^*}H^{-1}(g - g_C^{EI}\nu^*)
\tag{23}
$$

where $\lambda$ and $\nu$ are the Lagrangian multipliers of the KL-divergence term and the constraint term in the Lagrangian function, respectively. $\lambda^*, \nu^*$ are the solutions to the dual problem:

$$
\nu^* = \max\{0, \frac{\lambda^* c - u}{v}\}
\tag{24}
$$

$$
\lambda^* = \arg\max_{\lambda \geq 0}
\begin{cases}
\frac{1}{2\lambda}\left(\frac{u^2}{v} - q\right) + \frac{\lambda}{2}\left(\frac{c^2}{v} - \varphi\right) - \frac{uc}{v}, & \text{if} \lambda c > u \\
-\frac{1}{2}\left(\frac{q}{\lambda} + \lambda\varphi\right), & \text{otherwise,}
\end{cases}
\tag{25}
$$

where $q = g^T H^{-1} g$, $u = g^T H^{-1} g_C^{EI}$, $v = (g_C^{EI})^T H^{-1} g_C^{EI}$.

When the constraint is violated, we use the conjugate gradient method to decrease the constraint value:

$$
\theta^* = \theta_k - \left(\frac{2\varphi}{(g_C^{EI})^T H^{-1} g_C^{EI}}\right)^{\frac{1}{2}} H^{-1} g_C^{EI}
\tag{26}
$$

### A.2. MICE-PIDLag

In the MICE-PIDLag method, we write the CRL problem 20 as the first-order dynamical system:

$$
\begin{aligned}
\theta_{k+1} &= \theta_k + \eta(g - \lambda_k g_C^{EI}) \\
y_k &= J_C(\pi_k) + \frac{1}{1-\gamma}\mathbb{E}_{s\sim d^\pi, a\sim\pi}[A_C^{EI}(s, a|\pi_k)] \\
\lambda_k &= h(y_0, \cdots, y_k, d)
\end{aligned}
\tag{27}
$$

where $\eta$ is the step size of the update, $g$ is the gradient of reward objective in equation 20 and $g_C^{EI}$ denotes the gradient of extrinsic-intrinsic constraint in equation 20. $h$ denotes the control function. $\lambda$ is the Lagrangian multiplier for the equation 20. We provide the updated formulas for the Lagrangian multiplier in MICE-PIDLag:

$$
\lambda \leftarrow (K_P\Delta + K_I I + K_D\partial)_+
\tag{28}
$$

where $(\cdot)_+ = \max\{0, \cdot\}$, and:

$$
\begin{aligned}
\Delta &\leftarrow (J_C(\pi_k) + \frac{1}{1-\gamma}\mathbb{E}_{s\sim d^\pi, a\sim\pi}[A_C^{EI}(s, a|\pi_k)] - d), \\
I &\leftarrow (I + \Delta)_+, \\
\partial &\leftarrow \left(J_C(\pi_k) + \frac{1}{1-\gamma}\mathbb{E}_{s\sim d^\pi, a\sim\pi}[A_C^{EI}(s, a|\pi_k)] - J_C(\pi_{k-1}) - \frac{1}{1-\gamma}\mathbb{E}_{s\sim d^\pi, a\sim\pi}[A_C^{EI}(s, a|\pi_{k-1})]\right)_+ \\
&= \frac{1}{1-\gamma}\left(\mathbb{E}_{s\sim d^\pi, a\sim\pi}[A_C^{EI}(s, a|\pi_k)] - \mathbb{E}_{s\sim d^\pi, a\sim\pi}[A_C^{EI}(s, a|\pi_{k-1})] + \mathbb{E}_{s\sim d^{\pi_k}, a\sim\pi_k}[A_C^{\pi_{k-1}}(s, a)]\right)_+
\end{aligned}
\tag{29}
$$

$K_P$, $K_I$, and $K_D$ are the coefficients of the respective control terms. The initial value of the integral term $I$ is 0.

## B. Theoretical Proof

### B.1. Estimation Bias Lemma

**Lemma B.1.** *In a finite MDP for a given state-action pair $(s, a)$, the difference between the optimal cost value function $Q_C^*(s, a)$ and the cost value estimate $Q_{C,m}^{EI}(s, a)$ after $(m + n)$-th update is given by:*

$$Q_C^*(s, a) - Q_{C,n+m}^{EI}(s, a) = (1 - \alpha)^m[Q_C^*(s, a) - Q_{C,n}^{EI}(s, a)] - \alpha \sum_{i=1}^{m}(1 - \alpha)^{i-1}t_{n+m-i}(s, a) \tag{30}$$

*where $Q_{C,n}^{EI}(s, a)$ is the estimate of the value function at the $n$-th update, $\alpha$ is the step size, and $t_n(s, a) = c^E + \beta c^I + \gamma \mathbb{E}_{s'}[Q_{C,n}^{EI}(s', a')] - Q_C^*(s, a)$ is the target difference.*

*Proof.* We use induction method to proof this lemma B.1.

Base Case: m = 1

Substituting $m = 1$ in lemma B.1, we get:

$$Q_C^*(s, a) - Q_{C,n+1}^{EI}(s, a) = (1 - \alpha)[Q_C^*(s, a) - Q_{C,n}^{EI}(s, a)] - \alpha t_n(s, a) \tag{31}$$

According to the update equation of the extrinsic-intrinsic cost value function in MICE, we get:

$$\begin{aligned} Q_{C,n+1}^{EI}(s, a) &= (1 - \alpha)Q_{C,n}^{EI}(s, a) + \alpha(c^E + \beta c^I + \gamma \mathbb{E}_{s'}[Q_{C,n}^{EI}(s', a')]) \\ &= (1 - \alpha)Q_{C,n}^{EI}(s, a) + \alpha(t_n(s, a) + Q_C^*(s, a)) \end{aligned} \tag{32}$$

which is equivalent to equation 31.

Induction Step: $m = k + 1$

Assuming lemma B.1 is true for $m = k$, which is:

$$Q_C^*(s, a) - Q_{C,n+k}^{EI}(s, a) = (1 - \alpha)^k[Q_C^*(s, a) - Q_{C,n}^{EI}(s, a)] - \alpha \sum_{i=1}^{k}(1 - \alpha)^{i-1}t_{n+k-i}(s, a) \tag{33}$$

Now we need to prove that it holds for $m = k + 1$. According to the cost value update equation in MICE, we get:

$$\begin{aligned} Q_{C,n+k+1}^{EI}(s, a) &= (1 - \alpha)Q_{C,n+k}^{EI}(s, a) + \alpha(c^E + \beta c^I + \gamma \mathbb{E}_{s'}[Q_{C,n+k}^{EI}(s', a')]) \\ &= (1 - \alpha)Q_{C,n+k}^{EI}(s, a) + \alpha(t_{n+k}(s, a) + Q_C^*(s, a)) \end{aligned} \tag{34}$$

Then we can get:

$$Q_C^*(s, a) - Q_{C,n+k+1}^{EI}(s, a) = (1 - \alpha)[Q_C^*(s, a) - Q_{C,n+k}^{EI}(s, a)] - \alpha t_{n+k}(s, a) \tag{35}$$

Substituting the equation 33, we get:

$$\begin{aligned} &Q_C^*(s, a) - Q_{C,n+k+1}^{EI}(s, a) \\ =&(1 - \alpha)\left[(1 - \alpha)^k[Q_C^*(s, a) - Q_{C,n}^{EI}(s, a)] - \alpha \sum_{i=1}^{k}(1 - \alpha)^{i-1}t_{n+k-i}(s, a)\right] - \alpha t_{n+k}(s, a) \\ =&(1 - \alpha)^{k+1}[Q_C^*(s, a) - Q_{C,n}^{EI}(s, a)] - \alpha \sum_{i=1}^{k+1}(1 - \alpha)^{i-1}t_{n+k+1-i}(s, a) \end{aligned} \tag{36}$$

which satisfies the equation when $m = k + 1$ in lemma B.1.

□

Lemma B.1 indicates that when $m = 1$ and $Q_{C,n}^{EI}(s, a)$ is a random initial value for the cost value function, in a stochastic high value region of the state-action space, it is likely that $Q_C^*(s, a) > Q_{C,n}^{EI}(s, a)$ (Karimpanal et al., 2023). In this case, overestimation of our extrinsic-intrinsic cost value function can effectively reduce the estimation bias, whereas underestimation in the traditional value function further increases the estimation bias.

## B.2. Balancing Intrinsic Factor

**Proposition B.2.** *For a transition $(s, a, c^E, s')$ in a CMDP, where the $n$-th update of the extrinsic-intrinsic cost value $Q_n$ corresponds to the $n$-th update target $Q_{T_n}$, the modified target for the $(n + 1)$-th update is $Q'_{T_{n+1}}$, the balancing intrinsic factor $\beta'$ for the $(n + 1)$-th update is given by:*

$$\beta' = max\{\gamma^n(\beta_n - \frac{\alpha\epsilon_n}{c^I}), 0\}, \quad c^I > 0 \tag{37}$$

*where $\epsilon_n = Q_n - Q^*$ is the $n$-th update estimation bias, $\gamma \in (0, 1)$ is the discount factor.*

To avoid excessive estimate bias, we propose an adaptive mechanism for bias correction. The target for the $(n + 1)$-th update $Q_{T_{n+1}}$ is modified as:

$$Q'_{T_{n+1}} = Q_{T_{n+1}} - \alpha(Q_n - Q^*) \tag{38}$$

where $Q^*$ is the optimal value, $Q_n$ is the $n$-th update estimate. When $Q_n - Q^* < 0$, the modified target $Q'_{T_{n+1}}$ is adjusted upward, introducing positive bias to mitigate excessive underestimation. Conversely, when $Q_n - Q^* > 0$, $Q'_{T_{n+1}}$ is adjusted downward, introducing negative bias to address excessive overestimation. (Karimpanal et al., 2023).

Denote $\epsilon_n = Q_n - Q^*$ as the estimation bias of the $n$-th update.

$$\begin{aligned} Q'_{T_{n+1}} &= Q_{T_{n+1}} - \alpha(Q_n - Q^*) \\ &= Q_{T_{n+1}} - \alpha\epsilon_n \end{aligned} \tag{39}$$

The ideal estimation bias for the ideal $(n + 1)$-th update target can be denoted as:

$$\epsilon'_{n+1} = Q'_{T_{n+1}} - Q^* \tag{40}$$

According to equation 39, the ideal estimation bias for the $(n + 1)$-th target can be expressed as:

$$\epsilon'_{n+1} = Q_{T_{n+1}} - \alpha\epsilon_n - Q^* \tag{41}$$

where:

$$Q_{T_{n+1}} = c^E + \beta_n c^I + \gamma\mathbb{E}_{s'}[Q_{n+1}^{EI}(s', a')] \tag{42}$$

The we get the ideal estimation bias of the $(n + 1)$-th update, with the current value of balancing factor $\beta_n$:

$$\epsilon'_{n+1} = c^E + \beta_n c^I + \gamma\mathbb{E}_{s'}[Q_{n+1}^{EI}(s', a')] - \alpha\epsilon_n - Q^* \tag{43}$$

We define the ideal balancing factor $\beta'_{n+1}$ as $\beta'$:

$$\begin{aligned} \epsilon'_{n+1} &= Q'_{T_{n+1}} - Q^* \\ &= c^E + \beta' c^I + \gamma\mathbb{E}_{s'}[Q_{n+1}^{EI}(s', a')] - Q^* \end{aligned} \tag{44}$$

According to equation 43 and equation 44, we get the ideal factor $\beta'$:

$$\beta' = \beta_n - \frac{\alpha\epsilon_n}{c^I}, \quad c^I > 0 \tag{45}$$

To ensure the non-negativity of intrinsic cost, we clip the balancing factor:

$$\beta' = max\{\beta_n - \frac{\alpha\epsilon_n}{c^I}, 0\}, \quad c^I > 0 \tag{46}$$

Based on the convergence analysis of the traditional value function (Van Hasselt et al., 2016; Fujimoto et al., 2018), it gradually converges to the optimal value, causing the estimation bias to approach zero. To ensure the convergence of the extrinsic-intrinsic cost value function, we incorporate the discount factor to the balancing factor:

$$\beta' = max\{\gamma^n(\beta_n - \frac{\alpha\epsilon_n}{c^I}), 0\}, \quad c^I > 0 \tag{47}$$

We provide the convergence guarantee in Theorem 4.5.

$\square$

### B.3. Constraint Difference Lemma

**Lemma B.3.** *Given arbitrary two policies $\pi$ and $\pi'$, the difference in expectation constraint of extrinsic-intrinsic $J_C^{EI}(\pi')$ and extrinsic $J_C(\pi)$ can be expressed as:*

$$J_C^{EI}(\pi') - J_C(\pi) = \mathbb{E}_{\tau|\pi'}\left[\sum_{t=0}^{\infty}\gamma^t A_C^{EI}(s_t, a_t|\pi)\right] \tag{48}$$

*where $A_C^{EI}(s_t, a_t|\pi) = \mathbb{E}_{s_{t+1}}[c_t^E + \beta c_t^I + \gamma V_C^\pi(s_{t+1}) - V_C^\pi(s_t)]$. The expectation is taken over trajectories $\tau$, and $\mathbb{E}_{\tau|\pi'}$ indicates that actions are sampled from $\pi'$ to generate $\tau$.*

*Proof.* The expectations in $J_C^{EI}(\pi')$ and $J_C(\pi)$ can be expanded as:

$$J_C^{EI}(\pi') := \mathbb{E}_{\tau\sim\pi'}[\sum_{t=0}^{\infty}\gamma^t(c_t^E + \beta c_t^I)]$$
$$J_C(\pi) := \mathbb{E}_{\tau\sim\pi}[\sum_{t=0}^{\infty}\gamma^t c_t^E] = \mathbb{E}_{s_0\sim\rho}[V_C^\pi(s_0)] \tag{49}$$

$$
\begin{aligned}
&J_C^{EI}(\pi') - J_C(\pi) \\
=&\mathbb{E}_{\tau|\pi'}\left[\sum_{t=0}^{\infty}\gamma^t\left(c_t^E + \beta c_t^I\right)\right] - \mathbb{E}_{s_0\sim\rho}[V_C^\pi(s_0)] \\
=&\mathbb{E}_{\tau|\pi'}\left[\sum_{t=0}^{\infty}\gamma^t\left(c_t^E + \beta c_t^I\right) - V_C^\pi(s_0)\right] \\
=&\mathbb{E}_{\tau|\pi'}\left[\sum_{t=0}^{\infty}\gamma^t\left(c_t^E + \beta c_t^I + V_C^\pi(s_t) - V_C^\pi(s_t)\right)) - V_C^\pi(s_0)\right] \\
=&\mathbb{E}_{\tau|\pi'}\left[-V_C^\pi(s_0) + c_0^E + \beta c_0^I + V_C^\pi(s_0) - V_C^\pi(s_0) + \gamma c_1^E + \gamma\beta c_1^I + \gamma V_C^\pi(s_1) - \gamma V_C^\pi(s_1) + \cdots\right] \\
=&\mathbb{E}_{\tau|\pi'}\left[\sum_{t=0}^{\infty}\gamma^t\left(c_t^E + \beta c_t^I + \gamma V_C^\pi(s_{t+1}) - V_C^\pi(s_t)\right)\right] \\
=&\mathbb{E}_{\tau|\pi'}\left[\sum_{t=0}^{\infty}\gamma^t A_C^{EI}(s_t, \pi'(s_t)|\pi)\right]
\end{aligned} \tag{50}
$$

Here the term $A_C^{EI}(s_t, \pi'(s_t)|\pi)$ denotes that the advantage value function $A_C^{EI}$ is over $\pi$, the action is selected according to $\pi'$.

The second equation above holds because that

$$
\begin{aligned}
&\mathbb{E}_{\tau|\pi'}\left[V_C^\pi(s_0)\right] \\
=&\mathbb{E}_{s\sim d^{\pi'},a\sim\pi',s'\sim P}\left[V_C^\pi(s_0)\right] \\
=&\mathbb{E}_{s_0\sim\rho}\left[V_C^\pi(s_0)\right]
\end{aligned}
\tag{51}
$$

The initial state $s_0$ in $V_C^\pi(s_0)$ depends solely on the initial state distribution $\rho$, allowing the expectation over $\tau|\pi'$ to be expressed as an expectation over $s_0 \sim \rho$.

The third equation in equation 50 holds by adding $V_C^\pi$ while subtracting $V_C^\pi$. The fourth equation expands the cumulative sum over time steps $t$. The final equation follows from the definition of $A_C^{EI}$.

$\square$

## B.4. Constraint Bounds

**Theorem B.4** (Extrinsic-intrinsic Constraint Bounds). *For arbitrary two policies $\pi'$ and $\pi$, the following bound for the expected extrinsic-intrinsic constraint holds:*

$$
J_C^{EI}(\pi') - J_C^{EI}(\pi) \le \frac{1}{1-\gamma}\mathbb{E}_{s\sim d^\pi,a\sim\pi'}\left[A_C^{EI}(s,a|\pi) + \frac{2\gamma\epsilon_{\pi'}^{EI}}{1-\gamma}D_{TV}(\pi'\|\pi)[s]\right]
\tag{52}
$$

*where $\epsilon_{\pi'}^{EI} := \max_s |\mathbb{E}_{a\sim\pi'}[A_C^{EI}(s,a|\pi)]|$, the TV-divergence $D_{TV}(\pi'\|\pi)[s] = (1/2)\sum_a|\pi'(a|s) - \pi(a|s)|$.*

*Proof.* Define the state visit probability for time step $t$ as $p_\pi^t(s) = P(s_t = s|\pi)$, denote the transition matrix as $P_\pi(s'|s) = \int \mathrm{d}a\pi(a|s)P(s'|s,a)$, we get $p_\pi^t = P_\pi p_\pi^{t-1} = \cdots = P_\pi^t \rho$. The discounted future state distribution $d^\pi(s)$ satisfies:

$$
\begin{aligned}
d^\pi(s) &= (1-\gamma)\sum_{t=0}^\infty \gamma^t P(s_t = s|\pi) \\
&= (1-\gamma)\sum_{t=0}^\infty \gamma^t p_\pi^t(s) \\
&= (1-\gamma)\sum_{t=0}^\infty \gamma^t P_\pi^t \rho \\
&= (1-\gamma)\sum_{t=0}^\infty (\gamma P_\pi)^t \rho \\
&= (1-\gamma)(I - \gamma P_\pi)^{-1}\rho
\end{aligned}
\tag{53}
$$

where $\rho$ is the initial state distribution, $I$ is the identity matrix. Multiply both sides by $(I - \gamma P_\pi)$, we get

$$
(I - \gamma P_\pi)d^\pi(s) = (1-\gamma)\rho
\tag{54}
$$

For cost value function $V_C(s)$ with polices $\pi'$ and $\pi$, we get the following according to equation 54:

$$
\begin{aligned}
&(1-\gamma)\mathbb{E}_{s\sim\rho}[V_C(s)] + \mathbb{E}_{s\sim d^\pi,a\sim\pi,s'\sim P}[\gamma V_C(s')] - \mathbb{E}_{s\sim d^\pi}[V_C(s)] \\
=&(1-\gamma)\int \mathrm{d}s\rho(s)V_C(s) + \int \mathrm{d}s\int \mathrm{d}a\int \mathrm{d}s'd^\pi(s)\pi(a|s)P(s'|s,a)\gamma V_C(s') - \int \mathrm{d}sd^\pi(s)V_C(s) \\
=&\int \mathrm{d}s(1-\gamma)\rho(s)V_C(s) + \int \mathrm{d}sd^\pi(s)P_\pi\gamma V_C(s') - \int \mathrm{d}sd^\pi(s)V_C(s) \\
=&\int \mathrm{d}s(I - \gamma P_\pi)d^\pi(s)V_C(s) + \int \mathrm{d}sd^\pi(s)P_\pi\gamma V_C(s') - \int \mathrm{d}sd^\pi(s)V_C(s) \\
=&0
\end{aligned}
\tag{55}
$$

The third equation above holds according to equation 54. Then we get:

$$(1 - \gamma)\mathbb{E}_{s \sim \rho}[V_C(s)] + \mathbb{E}_{s \sim d^\pi, a \sim \pi, s' \sim P}[\gamma V_C(s')] - \mathbb{E}_{s \sim d^\pi}[V_C(s)] = 0 \tag{56}$$

The definition of discount total extrinsic-intrinsic cost is:

$$J_C^{EI}(\pi) = \frac{1}{1 - \gamma}\mathbb{E}_{s \sim d^\pi, a \sim \pi, s' \sim P}[C^{EI}(s, a, s')] \tag{57}$$

By combining this with equation 56, we get the discount total extrinsic-intrinsic cost equation:

$$J_C^{EI}(\pi) = \mathbb{E}_{s \sim \rho}[V_C(s)] + \frac{1}{1 - \gamma}\mathbb{E}_{s \sim d^\pi, a \sim \pi, s' \sim P}[C^{EI}(s, a, s') + \gamma V_C(s') - V_C(s)] \tag{58}$$

where the first term on the right side is the estimate of the policy constraint, and the second term on the right side is the average extrinsic-intrinsic TD-error of the approximator. The extrinsic-intrinsic TD-error is: $\delta_V^{EI}(s, a, s') = C^{EI}(s, a, s') + \gamma V_C(s') - V_C(s)$. According to the equation 58, the expectation extrinsic-intrinsic constraint difference of any two policies is:

$$J_C^{EI}(\pi') - J_C^{EI}(\pi) = \frac{1}{1 - \gamma}\left(\mathbb{E}_{s \sim d^{\pi'}, a \sim \pi', s' \sim P}[\delta_V^{EI}(s, a, s')] - \mathbb{E}_{s \sim d^\pi, a \sim \pi, s' \sim P}[\delta_V^{EI}(s, a, s')]\right) \tag{59}$$

To simplify the representation, we denote $\bar{\delta}_{\pi'}(s) = \mathbb{E}_{a \sim \pi', s' \sim P}[\delta_V^{EI}(s, a, s')]$. The first term of the right side in equation 59 can be represented as:

$$\begin{aligned}
\mathbb{E}_{s \sim d^{\pi'}, a \sim \pi', s' \sim P}[\delta_V^{EI}(s, a, s')] &= \int \mathrm{d}s d^{\pi'} \int \mathrm{d}a \pi' \int \mathrm{d}s' P \delta_V^{EI}(s, a, s') \\
&= \langle d^{\pi'}, \bar{\delta}_{\pi'} \rangle \\
&= \langle d^\pi, \bar{\delta}_{\pi'} \rangle + \langle d^{\pi'} - d^\pi, \bar{\delta}_{\pi'} \rangle
\end{aligned} \tag{60}$$

the second equation holds by adding $d^\pi$ while subtracting $d^\pi$.

According to the Hölder's inequality, for any $p, q \in [1, \infty]$ satisfy $\frac{1}{p} + \frac{1}{q} = 1$, we set $p = 1$ and $q = \infty$, and get:

$$\langle d^{\pi'} - d^\pi, \bar{\delta}_{\pi'} \rangle \leq \|d^{\pi'} - d^\pi\|_1 \|\bar{\delta}_{\pi'}\|_\infty \tag{61}$$

According to the definition in this theorem, we have $\|\bar{\delta}_{\pi'}\|_\infty = \epsilon_{\pi'}^{EI}$, and $\|d^{\pi'} - d^\pi\|_1 = 2D_{TV}(d^{\pi'}\|d^\pi)$. According to Lemma 3 in CPO, we have:

$$\|d^{\pi'} - d^\pi\|_1 \leq \frac{2\gamma}{1 - \gamma}\mathbb{E}_{s \sim d^\pi}[D_{TV}(\pi'\|\pi)[s]] \tag{62}$$

By the importance sampling, we get:

$$\langle d^\pi, \bar{\delta}_{\pi'} \rangle = \langle \frac{\pi'}{\pi} d^\pi, \bar{\delta}_\pi \rangle \tag{63}$$

The second term of the right side in equation 59 can be represented as:

$$\begin{aligned}
\mathbb{E}_{s \sim d^\pi, a \sim \pi, s' \sim P}[\delta_V^{EI}(s, a, s')] &= \int \mathrm{d}s d^\pi \int \mathrm{d}a \pi \int \mathrm{d}s' P \delta_V^{EI}(s, a, s') \\
&= \langle d^\pi, \bar{\delta}_\pi \rangle
\end{aligned} \tag{64}$$

Then we get the final result by combining the above equations:

$$
\begin{aligned}
J_C^{EI}(\pi') - J_C^{EI}(\pi) &\le \frac{1}{1-\gamma}\left(\langle\frac{\pi'}{\pi}d^\pi, \bar{\delta}_\pi\rangle + 2D_{TV}(d^{\pi'}\|d^\pi)\epsilon_{\pi'}^{EI} - \langle d^\pi, \bar{\delta}_\pi\rangle\right) \\
&= \frac{1}{1-\gamma}\left((\frac{\pi'}{\pi}-1)\langle d^\pi, \bar{\delta}_\pi\rangle + 2D_{TV}(d^{\pi'}\|d^\pi)\epsilon_{\pi'}^{EI}\right) \\
&\le \frac{1}{1-\gamma}\mathbb{E}_{s\sim d^\pi, a\sim\pi, s'\sim P}\left[(\frac{\pi'}{\pi}-1)\delta_V^{EI}(s,a,s') + \frac{2\gamma\epsilon_{\pi'}^{EI}}{1-\gamma}D_{TV}(\pi'\|\pi)[s]\right] \\
&= \frac{1}{1-\gamma}\mathbb{E}_{s\sim d^\pi, a\sim\pi'}\left[A_C^{EI}(s,a|\pi) + \frac{2\gamma\epsilon_{\pi'}^{EI}}{1-\gamma}D_{TV}(\pi'\|\pi)[s]\right]
\end{aligned}
\tag{65}
$$

$\square$

**Theorem B.5** (MICE Update Worst-Case Constraint Violation). *Suppose $\pi_k$, $\pi_{k+1}$ are related by the optimization objective 21, an upper bound on the constraint of the updated policy $\pi_{k+1}$ is:*

$$
J_C(\pi_{k+1}) \le d - I + \frac{\sqrt{2\varphi}\gamma\epsilon_C^{\pi_{k+1}}}{(1-\gamma)^2}
\tag{66}
$$

*where $\epsilon_C^{\pi_{k+1}} := \max_s |\mathbb{E}_{a\sim\pi_{k+1}}[A_C^{\pi_k}(s,a)]|$, and $I = \mathbb{E}_{\tau|\pi_{k+1}}\left[\sum_{t=0}^\infty \gamma^t\beta c_t^I\right]$.*

*Proof.* According to Corollary 2 in CPO (Achiam et al., 2017),

$$
J_C(\pi_{k+1}) - J_C(\pi_k) \le \frac{1}{1-\gamma}\mathbb{E}_{s\sim d^{\pi_k}, a\sim\pi_{k+1}}\left[A_C^{\pi_k}(s,a) + \frac{2\gamma\epsilon_C^{\pi_{k+1}}}{1-\gamma}D_{TV}(\pi_{k+1}\|\pi_k)[s]\right]
\tag{67}
$$

As $\pi_k$, $\pi_{k+1}$ are related by objective 21, we get

$$
J_C(\pi_k) + \frac{1}{1-\gamma}\mathbb{E}_{s\sim d^{\pi_k}, a\sim\pi_{k+1}}[A_C^{EI}(s,a|\pi_k)] \le d
\tag{68}
$$

which is:

$$
\begin{aligned}
J_C(\pi_k) + \frac{1}{1-\gamma}\mathbb{E}_{s\sim d^{\pi_k}, a\sim\pi_{k+1}}[c^E + \beta c^I + \gamma V_C(s') - V_C(s)] &\le d \\
J_C(\pi_k) + \frac{1}{1-\gamma}\mathbb{E}_{s\sim d^{\pi_k}, a\sim\pi_{k+1}}[A_C^{\pi_k}(s,a)] + \frac{1}{1-\gamma}\mathbb{E}_{s\sim d^{\pi_k}, a\sim\pi_{k+1}}[\beta c^I] &\le d \\
J_C(\pi_k) + \frac{1}{1-\gamma}\mathbb{E}_{s\sim d^{\pi_k}, a\sim\pi_{k+1}}[A_C^{\pi_k}(s,a)] &\le d - I
\end{aligned}
\tag{69}
$$

According to Pinsker's inequality, for arbitrary distributions $p$, $q$, the TV-divergence and KL-divergence are related by:

$$
D_{TV}(p\|q) \le \sqrt{\frac{D_{KL}(p\|q)}{2}}
\tag{70}
$$

According to Jensen's inequality, we get:

$$
\begin{aligned}
\mathbb{E}_{s\sim d^{\pi_k}}[D_{TV}(\pi_{k+1}\|\pi_k)[s]] &\le \sqrt{\frac{1}{2}\mathbb{E}_{s\sim d^{\pi_k}}[D_{KL}(\pi_{k+1}\|\pi_k)[s]]} \\
&\le \sqrt{\frac{\varphi}{2}}
\end{aligned}
\tag{71}
$$

Then we get the final result:

$$
J_C(\pi_{k+1}) \le d - I + \frac{\sqrt{2\varphi}\gamma\epsilon_C^{\pi_{k+1}}}{(1-\gamma)^2}
\tag{72}
$$

□

## B.5. Convergence Analysis

Based on the same assumptions as in TD3 and Double Q-learning, we give convergence guarantees of the extrinsic-intrinsic cost value function in MICE.

**Lemma B.6.** *Consider a stochastic process* $(\zeta_t, \Delta_t, F_t), t \geq 0$ *where* $\zeta_t, \Delta_t, F_t : X \to \mathbb{R}$ *satisfy the equation:*

$$\zeta_{t+1}(x_t) = (1 - \zeta_t(x_t))\Delta_t(x_t) + \zeta_t(x_t)F_t(x_t) \tag{73}$$

*where* $x_t \in X$ *ant* $t = 0, 1, 2, \cdots$. *Let* $P_t$ *be a sequence of increasing* $\sigma$-*fields such that* $\zeta_0$ *and* $\Delta_0$ *are* $P_0$-*measurable and* $\zeta_t$, $\Delta_t$ *and* $F_{t-1}$ *are* $P_t$-*measurable,* $t = 1, 2, \cdots$. *Assume that the following holds:*

1. *The set* $X$ *is finite.*

2. $\zeta_t(x_t) \in [0, 1]$, $\sum_t \zeta_t(x_t) = \infty$, $\sum_t (\zeta_t(x_t))^2 < \infty$ *with probability 1 and* $\forall x \neq x_t : \zeta(x) = 0$.

3. $\| \mathbb{E}[F_t | P_t] \|_\infty \leq \kappa \| \Delta_t \|_\infty + c_t$ *where* $\kappa \in [0, 1)$ *and* $c_t$ *converges to 0 with probability 1.*

4. $Var[F_t(x_t)|P_t] \leq K(1 + \kappa \| \Delta_t \|_\infty)^2$, *where* $K$ *is some constant.*

*where* $\| \cdot \|_\infty$ *denotes the maximum norm. Then* $\Delta_t$ *converges to 0 with probability 1.*

We use the Lemma B.6 to prove the convergence of our approach with a similar condition in Q-learning.

**Theorem B.7** (Convergence Analysis). *Given the following conditions:*

1. *Each state-action pair is sampled an infinite number of times.*

2. *The MDP is finite.*

3. $\gamma \in [0, 1)$.

4. $Q_C$ *values are stored in a lookup table.*

5. $Q_C$ *receives an infinite number of updates.*

6. *The learning rates satisfy* $\alpha_t(s, a) \in [0, 1]$, $\sum_t \alpha_t(s, a) = \infty$, $\sum_t (\alpha_t(s, a))^2 < \infty$ *with probability 1 and* $\alpha_t(s, a) = 0$, $\forall (s, a) \neq (s_t, a_t)$.

7. $Var[c_t^E + \beta c_t^I] < \infty, \forall s, a$.

*The extrinsic-intrinsic* $Q_C^{EI}$ *will converge to the optimal value function* $Q_C^*$ *with probability 1.*

Theorem B.7 ensures that our method converges to the optimal solution.

*Proof.* We apply Lemma B.6 to prove Theorem B.7. Denote the variables in Lemma B.6 with $P_t = \{Q_{C0}^{EI}, s_0, a_0, \alpha_0, c_1^E, s_1, \cdots, s_t, a_t\}$, $X = S \times A$, $\zeta_t = \alpha_t$. Define $\Delta_t(s_t, a_t) = Q_{Ct}^{EI}(s_t, a_t) - Q_C^*(s_t, a_t)$, $F_t = c_t^E + \beta c_t^I + \gamma Q_{Ct}^{EI}(s_{t+1}, a_{t+1}) - Q_C^*(s_t, a_t)$.

Condition 1 of the lemma B.6 holds by condition 2 of the theorem B.7. Condition 2 of the lemma B.6 holds as the theorem condition 6 with $\zeta_t = \alpha_t$. The condition 4 of lemma B.6 holds as a consequence of the condition 7 in the theorem.

So we need to show that the lemma condition 3 on the expected contraction of $F_t$ holds.

The extrinsic-intrinsic Q-learning equation in our paper is:

$$Q_C^{EI}(s, a) = (1 - \alpha)Q_C^{EI}(s, a) + \alpha(c^E + \beta c^I + \gamma \mathbb{E}_{s'}[Q_C^{EI}(s', a')]) \tag{74}$$

We have:

$$
\begin{aligned}
\Delta_{t+1}(s_t, a_t) =& Q^{EI}_{Ct+1}(s_t, a_t) - Q^*_C(s_t, a_t) \\
=& (1 - \alpha_t)Q^{EI}_{Ct}(s_t, a_t) + \alpha_t(c^E_t + \beta c^I_t + \gamma Q^{EI}_t(s_{t+1}, a_{t+1})) - Q^*_C(s_t, a_t) \\
=& (1 - \alpha_t)(Q^{EI}_{Ct}(s_t, a_t) - Q^*_C(s_t, a_t)) + \alpha_t(c^E_t + \beta c^I_t + \gamma Q^{EI}_{Ct}(s_{t+1}, a_{t+1}) - Q^*_C(s_t, a_t)) \\
=& (1 - \alpha_t)\Delta_t + \alpha_t F_t
\end{aligned}
\tag{75}
$$

For the $F_t$, we can write as:

$$
\begin{aligned}
F_t(s_t, a_t) =& c^E_t + \beta c^I_t + \gamma Q^{EI}_{Ct}(s_{t+1}, a_{t+1}) - Q^*(s_t, a_t) \\
=& F^E_t(s_t, a_t) + \beta c^I_t
\end{aligned}
\tag{76}
$$

where $F^E_t(s_t, a_t) = c^E_t + \gamma Q^{EI}_{Ct}(s_{t+1}, a_{t+1}) - Q^*_C(s_t, a_t)$ is the value of $F_t$ in normal value function. According to the convergence analysis in value function, we get $\mathbb{E}[F^E_t | P_t] \leq \gamma \parallel \Delta_t \parallel_\infty$. Then condition 3 of lemma 2 holds if $c^I_t$ converges to 0 with probability 1.

The intrinsic cost is normalized and the discount factor term $\gamma^n$ is included in the balancing factor, where $\gamma \sim (0, 1)$. So $\beta c^I_t$ converges to 0 with probability 1, which then shows condition 3 of lemma B.6 is satisfied. So the $Q^{EI}_C(s_t, a_t)$ converges to $Q^*(s_t, a_t)$.

$\square$

## C. Experiment

### C.1. Algorithm Process

We provide the code for MICE-CPO and MICE-PIDLag in https://github.com/ShiqingGao/MICE. A formal description of our method is shown in Algorithm 1.

---

**Algorithm 1** MICE: Memory-driven Intrinsic Cost Estimation

---

**Input:** Initialize policy network $\pi_\theta$, value networks $V^\omega_R$ and $V^\psi_C$, flashbulb memory $M$. Set the hyperparameter.
**Output:** The optimal policy parameter $\theta$.
1: **for** epoch k=0,1,2,... **do**
2:     Sample under the current policy $\pi_{\theta_k}$.
3:     Update flashbulb memory $M$.
4:     Output the intrinsic cost $c^I$.
5:     Process the trajectories to $C$-returns, calculate extrinsic-intrinsic advantage functions $A^{EI}$ with $V^\psi_C$ and $c^I$ by GAE method.
6:     **for** K iterations **do**
7:         Update value networks $V^\omega_R, V^\psi_C$.
8:         Update policy network $\pi_\theta$.
9:         **if** $\frac{1}{N}\sum_{j=1}^N D_{KL}(\pi_\theta || \pi_{\theta_k})[s_j] > \varphi$ **then**
10:           Break.
11:         **end if**
12:     **end for**
13: **end for**
14: **Return:** Policy parameters $\theta = \theta_{k+1}$.

---

### C.2. Complexity Analysis

The Time complexity of MICE is $O(M \cdot N)$, where $M$ represents the size of the flashbulb memory, and $N$ denotes the number of samples processed during training. For the state of each sample in the training, we compute the distance between the embedding generated by the random projection layer and the contents of the memory in order to retrieve the $k$-nearest neighbors.

## C.3. Additional Experiments

We conducted comparative experiments against additional baselines. Additionally, we designed ablation study to verify the effect of different components in MICE.

### C.3.1. BASELINES

The MICE approach can be extended to other CRL algorithms based on actor-critic architectures. We compare MICE-CPO and MICE-PIDLag with their corresponding baselines, CPO and PIDLag, across multiple environments to validate the improvements offered by our approach. The results are shown in Figure 8 and Figure 9, which indicates that our MICE approach effectively improves constraint satisfaction over the respective original approaches while maintaining the same or even better level of policy performance. It is crucial to note that in SafetyCarGoal1-v0, CPO exceeds the constraint threshold, so its higher return compared to MICE-CPO does not indicate a better policy. A direct comparison of the returns of CPO and MICE-CPO is not meaningful. A similar phenomenon occurs in SafetyAntVelocity-v4 when evaluating MICEPID and PIDLag.

Here, we provide an additional introduction to the baseline methods employed in the main text. CUP (Yang et al., 2022) is a projection approach that provides generalized theoretical guarantees for surrogate functions with a generalized advantage estimator (Schulman et al., 2015), effectively reducing variance while maintaining acceptable bias. State augmentation methods aim to achieve constraint satisfaction with probability one. Saute RL (Sootla et al., 2022a) eliminates safety constraints by expanding them into the state space and reshaping the objective. Specifically, the residual safety budget is treated as a new state to quantify the risk of violating the constraint. Simmer (Sootla et al., 2022b) extends the state space with a state encapsulating the safety information. This safe state is initialized with a safety budget, and the value of the safe state can be used as a distance measure to the unsafe region. Simmer reduces safety constraint violations by scheduling the initial safety budget.

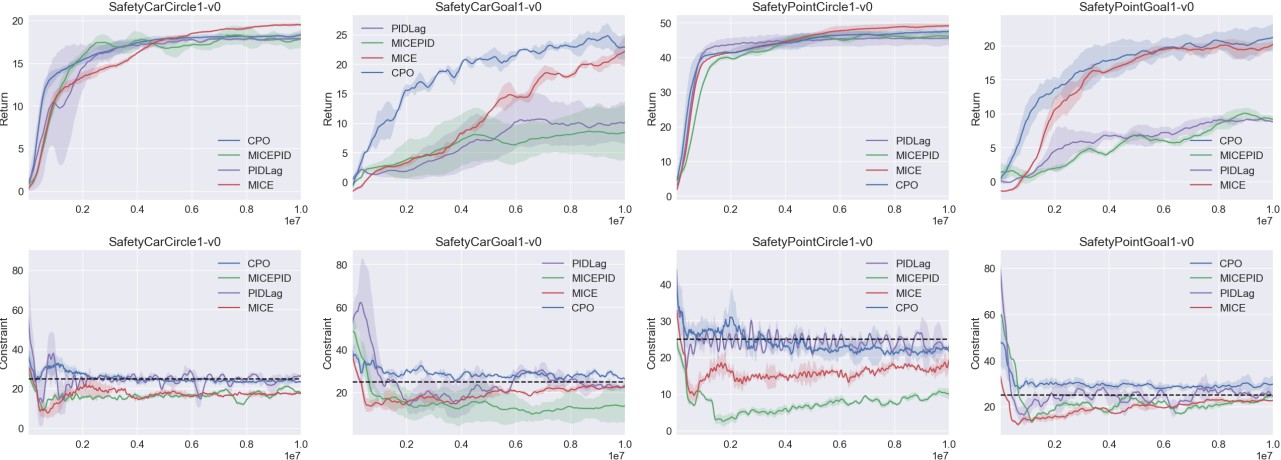

Figure 8: Comparison of MICE and their respective baseline approaches on Safety Gym. The x-axis is the total number of training steps, the y-axis is the average return or constraint. The solid line is the mean and the shaded area is the standard deviation. The dashed line in the cost plot is the constraint threshold which is 25.

### C.3.2. COMPLEX TASKS

We conducted additional experiments comparing MICE with CPO in the SafetyCarButton1-v0 and SafetyPointButton1-v0 environments. These tasks are more complex, requiring agents to navigate to a target button and correctly press it while avoiding Gremlins and Hazards. Results presented in Figure 10 show that MICE achieves superior constraint satisfaction while maintaining policy performance comparable to CPO.

### C.3.3. COMPARED TO TD3-BASED COST VALUE FUNCTION

TD3 is a reinforcement learning method designed to mitigate the overestimation bias in the reward value function, which uses the minimum output from two separately-learned action-value networks during policy updates. Similarly, TD3 can

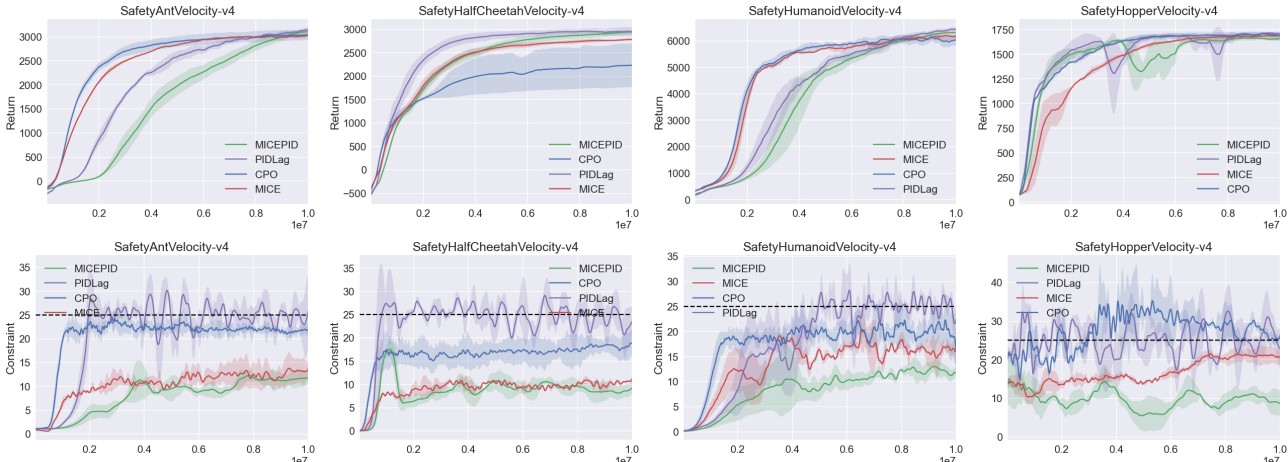

Figure 9: Comparison of MICE and their respective baseline approaches on Safety MuJoCo. The x-axis is the total number of training steps, the y-axis is the average return or constraint. The solid line is the mean and the shaded area is the standard deviation. The dashed line in the cost plot is the constraint threshold which is 25.

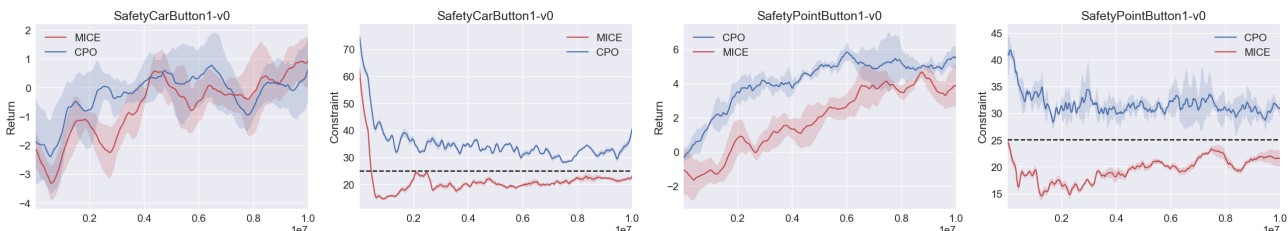

Figure 10: Comparison of MICE with CPO in more complex environments. MICE achieves better constraint satisfaction while maintaining policy performance comparable to CPO.

serve as a baseline for addressing underestimation bias in cost by using the maximum output from two separately-learned cost value networks. We conducted experiments to compare the cost value estimation bias between the TD3 cost value function and MICE with the PIDLag optimization method in SafetyPointGoal1-v0 and SafetyCarGoal1-v0, as shown in Figure 11, and the performance results are shown in Figure 12.

The results show that TD3 mitigates underestimation bias in cost value estimation, but it cannot fully eliminate it. This limitation arises from the inherent slow adaptation of neural networks, which results in a residual correlation between the value networks, thus preventing TD3 from completely eliminating the underestimation bias. In contrast, MICE eliminates this bias more accurately with the memory-driven intrinsic cost, resulting in significantly improved constraint satisfaction. The intrinsic cost in MICE enhances policy learning by enabling safer exploration, thereby improving performance while maintaining constraint compliance.

Additionally, compared to the TD3 cost value function, the flashbulb memory structures in MICE help address the catastrophic forgetting issue in neural networks (Lipton et al., 2016), where agents may forget previously encountered states and revisit them under new policies. This mechanism generates intrinsic cost signals that guide the agent away from previously explored dangerous states, effectively preventing repeated encounters with the same hazards.

### C.3.4. ABLATION STUDY ON RANDOM PROJECTION LAYER

In this paper, a random projection layer is used to compress states before applying KNN. Specifically, states are projected from their original dimension $n$ to a lower embedding dimension $m$ using a Gaussian random matrix of shape $(n, m)$. This reduces the computational complexity of KNN from $O(Mn)$ to $O(Mm)$, where $M$ is the number of stored states. According to the Johnson-Lindenstrauss lemma (Johnson et al., 1984), random projection approximately preserves Euclidean distances in the original space.

We further conducted ablation experiments on random projection, as shown in Figure 13, which confirm that using random

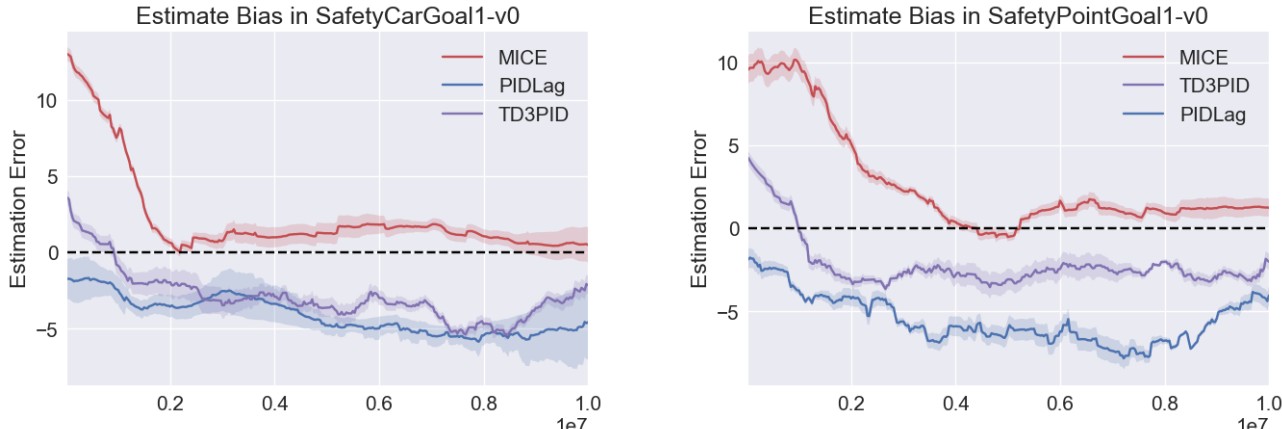

Figure 11: Comparison experiment about Estimation Error of MICE to TD3PID. The y-axis is the cost value estimate minus the true value, and the dashed line is the zero deviation. MICE achieves more accurate mitigation of estimation bias compared to TD3

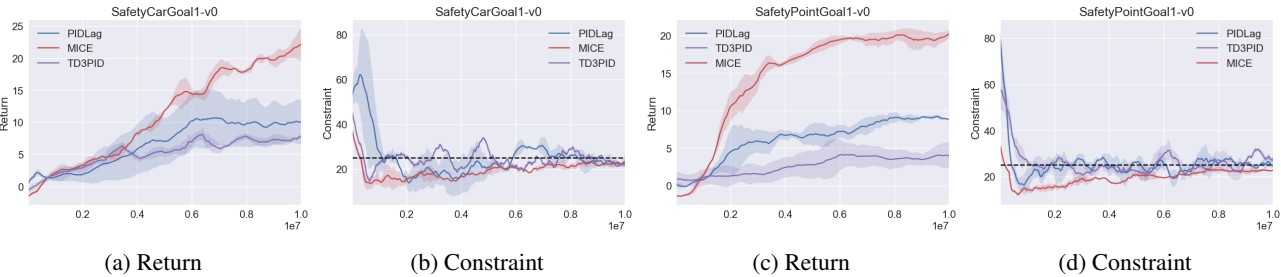

| (a) Return | (b) Constraint | (c) Return | (d) Constraint |

Figure 12: Comparison of MICE, TD3PID and PIDLag. The x-axis is the total number of training steps, the y-axis is the average return or constraint. The solid line is the mean and the shaded area is the standard deviation. The dashed line in the cost plot is the constraint threshold which is 25.

projection in MICE does not reduce policy performance or increase constraint violations, while substantially reducing training time.

In our work, relative similarity between states is more important than absolute scalar distances. In our implementation, both extrinsic and intrinsic costs are normalized to ensure training stability. This normalization alters absolute distance values but preserves similarities between states, which is key to maintaining meaningful guidance for intrinsic cost computation and policy learning. The ablation results show that random projection effectively preserves this relative similarity, thus maintaining the performance of MICE.

## C.4. Environments

### C.4.1. SAFETY GYM

Figure 14 shows the environments in the Safety Gym (Ji et al., 2023). Safety Gym is the standard API for safe reinforcement learning. The agent perceives the world through the sensors of the robots and interacts with the environment via its actuators in Safety Gym. In this work, we consider two agents, Point and Car, and two tasks, Goal and Circle.

The Point is a simple robot constrained to a two-dimensional plane. It is equipped with two actuators, one for rotation and another for forward/backward movement. It has a small square in front of it, making it easier to visually determine the orientation of the robot. The action space in Point consists of two dimensions ranging from -1 to 1, and the observation space consists of twelve dimensions ranging from negative infinity to positive infinity.

The Car is a more complex robot that moves in three-dimensional space and has two independently driven parallel wheels and a freely rotating rear wheel. For this robot, both steering and forward/backward movement require coordination between the two drive wheels. The action space of Car includes two dimensions with a range from -1 to 1, while the observation

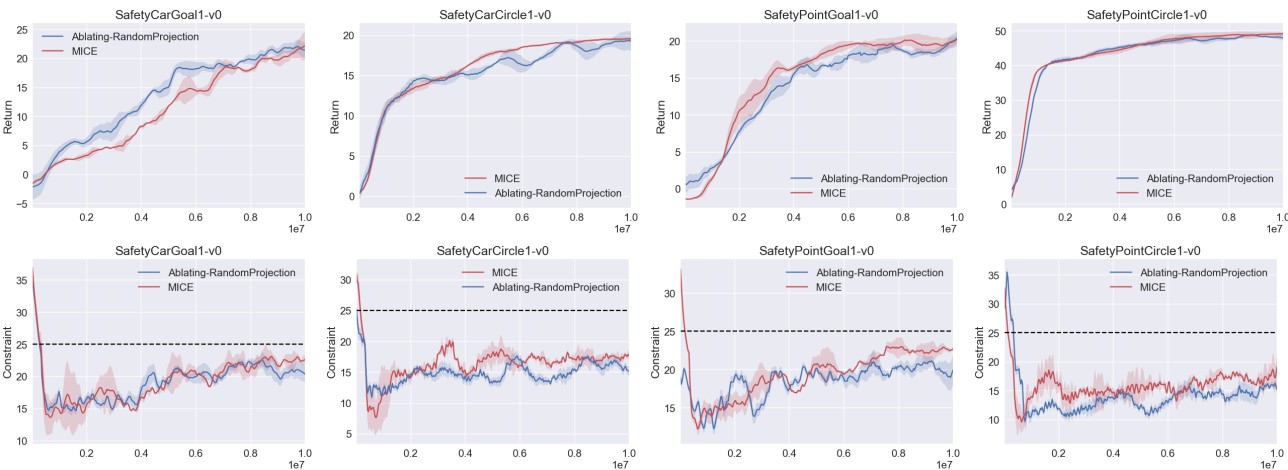

Figure 13: Ablation study of random projection layer in MICE algorithm. Utilizing random projection in MICE does not degrade policy performance or increase constraint violations.

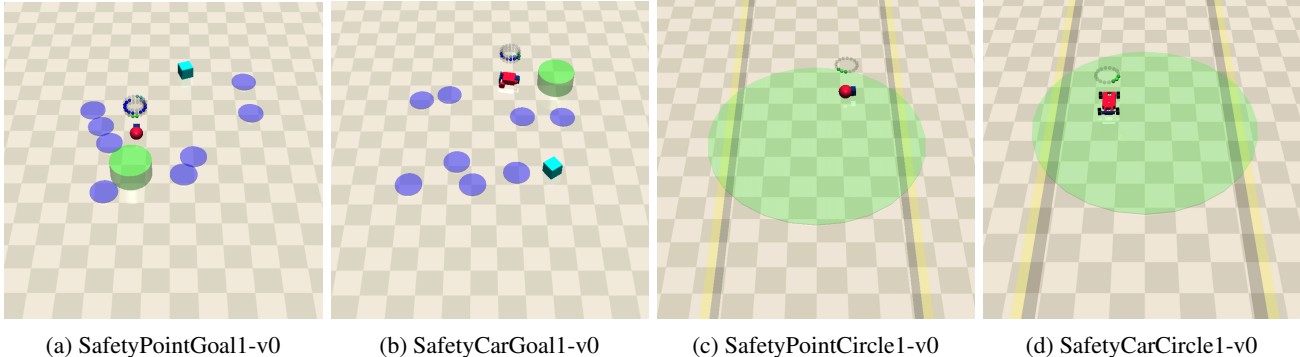

(a) SafetyPointGoal1-v0       (b) SafetyCarGoal1-v0       (c) SafetyPointCircle1-v0       (d) SafetyCarCircle1-v0

Figure 14: Environments in Safety Gym.

space consists of 24 dimensions with a range from negative infinity to positive infinity.

**Goal:** The agent is required to navigate towards the location of the goal. Upon successfully reaching the goal, the goal location is randomly reset to a new position while maintaining the remaining layout unchanged. The rewards in the task of Goal are composed of two components: reward distance and reward goal. In terms of reward distance, when the agent is closer to the Goal it gets a positive value of reward, and getting farther will cause a negative reward. Regarding the reward goal, each time the agent successfully reaches the Goal, it receives a positive reward value denoting the completion of the goal. In SafetyGoal1, the Agent needs to navigate to the Goal's location while circumventing Hazards. The environment consists of 8 Hazards positioned throughout the scene randomly.

**Circle:** Agent is required to navigate around the center of the circle area while avoiding going outside the boundaries. The optimal path is along the outermost circumference of the circle, where the agent can maximize its speed. The faster the agent travels, the higher the reward it accumulates. The episode automatically ends if the duration exceeds 500 time steps. When out of the boundary, the agent gets an activated cost.

### C.4.2. SAFETY MUJOCO

The agent in Safety MuJoCo is provided by OpenAI Gym (Brockman et al., 2016), and it is trained to move along a straight line while constrained with a velocity limit. Figure 15 illustrates the different environments.

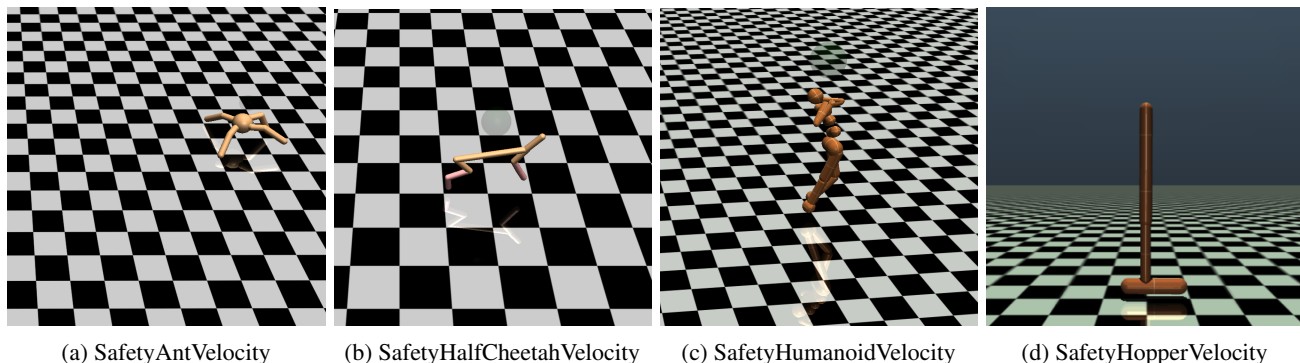

| (a) SafetyAntVelocity | (b) SafetyHalfCheetahVelocity | (c) SafetyHumanoidVelocity | (d) SafetyHopperVelocity |

Figure 15: Environments in Safety MuJoCo.

### C.5. HyperParameters

All experiments are implemented in Pytorch 2.0.0 and CUDA 11.3 and performed on Ubuntu 20.04.2 LTS with a single GPU (GeForce RTX 3090). The hyperparameters are summarized in Table 1.

| Parameter | Saute | Simmer | CUP | CPO | PIDLag | MICE-CPO | MICE-PIDLag |
|---|---|---|---|---|---|---|---|
| hidden layers | 2 | 2 | 2 | 2 | 2 | 2 | 2 |
| hidden sizes | 64 | 64 | 64 | 64 | 64 | 64 | 64 |
| activation | $tanh$ | $tanh$ | $tanh$ | $tanh$ | $tanh$ | $tanh$ | $tanh$ |
| actor learning rate | $3e-4$ | $3e-4$ | $3e-4$ | $3e-4$ | $3e-4$ | $3e-4$ | $3e-4$ |
| critic learning rate | $3e-4$ | $3e-4$ | $3e-4$ | $3e-4$ | $3e-4$ | $3e-4$ | $3e-4$ |
| batch size | 64 | 64 | 64 | 64 | 64 | 64 | 64 |
| trust region bound | $1e-2$ | $1e-2$ | $1e-2$ | $1e-2$ | $1e-2$ | $1e-2$ | $1e-2$ |
| discount factor gamma | 0.99 | 0.99 | 0.99 | 0.99 | 0.99 | 0.99 | 0.99 |
| GAE gamma | 0.95 | 0.95 | 0.95 | 0.95 | 0.95 | 0.95 | 0.95 |
| normalization coefficient | $1e-3$ | $1e-3$ | $1e-3$ | $1e-3$ | $1e-3$ | $1e-3$ | $1e-3$ |
| clip ratio | 0.2 | 0.2 | 0.2 | $N/A$ | 0.2 | $N/A$ | 0.2 |
| conjugate gradient damping | $N/A$ | $N/A$ | $N/A$ | 0.1 | $N/A$ | 0.1 | $N/A$ |
| initial lagrangian multiplier | $N/A$ | $N/A$ | $1e-3$ | $N/A$ | $1e-3$ | $N/A$ | $1e-3$ |
| lambda learning rate | $N/A$ | $N/A$ | 0.035 | $N/A$ | 0.035 | $N/A$ | 0.035 |

Table 1: Hyperparameters

