# OpenReview forum: "Controlling Underestimation Bias in Constrained Reinforcement Learning for Safe Exploration"
_ICML.cc/2025/Conference — ICML 2025 oral_

### Official Review · Reviewer_kehG · 2025-03-13

**Overall Recommendation:** 4

**Summary:**

This paper presents Memory-driven Intrinsic Cost Estimation (MICE), an algorithm for reducing constraint violations in constrained RL throughout training, rather than just at the end of training. It does this using a memory buffer that stores a representation previously seen constraint-violating states, which are compared to new states to form an intrinsic reward term to reduce value function over-estimation leading to repeated visitation of constraint violating states.

**Claims And Evidence:**

The claims of the paper seem to be well-supported by the theoretical and experimental evidence provided.

**Essential References Not Discussed:**

Conservative Q Learning and similar methods have been used for offline RL with success, while the goal and method are different the general concept of conservative value estimation to ensure good behavior is shared, it might be worth mentioning in the discussion of related work on overestimation in RL.

**Experimental Designs Or Analyses:**

The experimental design seems reasonable, the estimation bias and ablation experiments test specific claims the paper makes about why the proposed method works.

**Methods And Evaluation Criteria:**

The proposed methods and evaluation seems reasonable for the problem.

**Other Comments Or Suggestions:**

The paper title at the top of each page is not correct

**Other Strengths And Weaknesses:**

Overall, I liked this paper. The problem being addressed is significant and to my knowledge novel. The proposed method makes intuitive sense, and is theoretically justified, with sufficient experimental validation on standard benchmarks to demonstrate efficacy.

If there's one weakness to highlight, it's that I wonder about the use of a random network projection for the memory state comparison (I've got a question below along those lines), but overall I think this is a solid paper that is ready for acceptance.

**Questions For Authors:**

When using a random network to project into latent space, my understanding is that distance in random latent space is mostly a "these two states are or are not similar" thing, rather than having use as a scalar quantity. As MICE uses a k-NN distance in this space, it makes me wonder if the numerical value matters or if it's just a matter of being moderately distant from previously sampled datapoints. Could the intrinsic rewards be quantized, and would that affect performance? I don't think this is critical to answer but the use of a random latent space seems at odds with the idea of specific distances mattering.

**Relation To Broader Scientific Literature:**

This work seems significant in that reducing the frequency of constraint violations during constrained RL training is valuable for enabling CRL to be used for many tasks which require training where constraint violations have a real cost (for example, training a robot policy on a real robot). To my knowledge, prior work on CRL hasn't attempted to address during-training violations and has mostly focused on the state of the policy after training. That this work also improves the training process in general and finds a better solution is a worthwhile contribution as well.

**Theoretical Claims:**

I did not check the proofs, but the theorem claims seem reasonable

---

> ### Author Rebuttal · Authors · 2025-03-31
>
> We sincerely appreciate the reviewer's positive and insightful comments. The followings are detailed responses to the points raised by Reviewer kehG.
>
>
> >When using a random network to project into latent space, my understanding is that distance in random latent space is mostly a "these two states are or are not similar" thing, rather than having use as a scalar quantity. As MICE uses a k-NN distance in this space, it makes me wonder if the numerical value matters or if it's just a matter of being moderately distant from previously sampled datapoints. Could the intrinsic rewards be quantized, and would that affect performance? I don't think this is critical to answer but the use of a random latent space seems at odds with the idea of specific distances mattering.
>
> **Response:**
> We appreciate the reviewer’s insightful comments regarding the use of random projection in state comparisons.
>
> In our experiments, random projection is implemented using a Gaussian random matrix of shape $(n, m)$, which projects states from the original dimension $n$ to a lower dimension $m$. According to the Johnson-Lindenstrauss lemma [1], random projection approximately preserves Euclidean distances in the original space, which is proven to be valid in similar KNN-based methods in prior works [2][3].
>
> Additionally, we fully agree with the reviewer that relative similarity between states is more important than absolute scalar distance. In our implementation, both extrinsic and intrinsic costs are normalized to ensure training stability. This normalization alters absolute distance values but preserves similarities between states, which is key to maintaining meaningful guidance for intrinsic cost computation and policy learning.
>
> To further assess the impact of random projection, we conducted an additional ablation study on it. The results in Figure 1 (<https://anonymous.4open.science/r/7532-6C07/experiments3.pdf>) demonstrate that using random projection does not degrade policy performance or increase constraint violations, further supporting the idea that state similarity is likely more critical than absolute numerical values.
>
> We appreciate the reviewer’s constructive questions and hope these additional experiments and clarifications address the concern.
>
>
> >The paper title at the top of each page is not correct.
>
> **Response:**
> We sincerely appreciate the reviewer’s careful review. We will correct this typo in the revised manuscript.
>
>
> [1] Johnson, W. B., Lindenstrauss, J., et al. Extensions of lipschitz mappings into a hilbert space. Contemporary mathematics,
> 26(189-206):1, 1984.
>
> [2] Hu, H., Ye, J., Zhu, G., Ren, Z., and Zhang, C. Generalizable episodic memory for deep reinforcement learning. In
> International Conference on Machine Learning, pp. 4380–4390. PMLR, 2021.
>
> [3] Zhu, G., Lin, Z., Yang, G., and Zhang, C. Episodic reinforcement learning with associative memory. In International
> Conference on Learning Representations.

---

### Official Review · Reviewer_rbBW · 2025-03-14

**Overall Recommendation:** 4

**Summary:**

The paper proposed MICE, which introduces the concept of intrinsic cost to combat the issue of underestimation bias in cost-value function present in many safe RL algorithms. The paper discusses their flashbulb memory design, which attaches additional intrinsic cost to previously visited risk regions, and shows that this reduces underestimation bias in safe RL. Theoretically, the paper provided theoretical bound on the constraint violation and convergence guarantee.

----- Score raised to 4: Accept after rebuttal -----

**Claims And Evidence:**

Most of the claims in the paper are well supported by theory and experiment. The only doubt I have is that:

1. To compute the similarity between newly encountered state and the state in the memory, the paper proposed a kernel function which is approximated using k-nearest neighbors. KNN, as a non-parametric model, is known to suffer from curse of dimensionality. This might cause difficulty for MICE to handle very high-dimensional state representation. The embedding function $f()$ would be key in this regard. The paper might benefit from investigating how this embedding function interact with the effect of KNN choice.

**Essential References Not Discussed:**

NA

**Experimental Designs Or Analyses:**

1. In the 8 chosen tasks, MICE's performance seems rather close to CPO although it's relatively safer in two domains. I'd think that expanding to more tasks might verify that MICE is reliably safer than CPO.

2. In the sensitivity analysis (Appendix C.3.2) of discount factor, the oft-used default of 0.99 seems to produce a policy which oscillates around the constraint threshold 25. This behavior seems largely similar to other constrained RL algorithm (e.g. PPO-Lagrangian, PID-Lagrangian). The authors might want to consider expanding to other tasks to check how the discount rate of 0.99 behaves.

**Methods And Evaluation Criteria:**

The baselines chosen are well suited for this problem. I'd also highlight that MICE is extended based on CPO, thus the baseline comparison with CPO (in Appendix C.3.1) should be moved to the main paper. I do note that CPO and MICE performance are very close, and MICE is relatively safe in some domains (Hopper, PointGoal1).

**Other Comments Or Suggestions:**

1. Is discount rate of 0.99 used across all baselines for the main result?

**Other Strengths And Weaknesses:**

NA

**Questions For Authors:**

Please refer to earlier sections.

**Relation To Broader Scientific Literature:**

This is quite relevant to the safe RL literature where cost-value underestimation might be present.

**Theoretical Claims:**

The theoretical claim is well supported, I do have to highlight I did not do step-by-step check on the detailed proof in the Appendix.

---

> ### Author Rebuttal · Authors · 2025-03-31
>
> We sincerely appreciate the reviewer's positive and insightful comments. The following are the detailed responses to the points raised by Reviewer rbBW.
>
>
> >The paper might benefit from investigating how this embedding function interact with the effect of KNN choice.
>
> **Response:**
> We appreciate the reviewer’s valuable comment regarding investigating the embedding function and KNN choice.
>
> In our paper, we utilize a random projection method as the embedding function to compress states before applying KNN. Specifically, states are projected from their original dimension $n$ to a lower embedding dimension $m$ using a Gaussian random matrix of shape $(n, m)$. This significantly reduces the computational complexity of KNN from $O(N n)$ to $O(N m)$, where $N$ is the number of states in memory. Prior studies [1][2] have validated the effectiveness of random projection in similar KNN-based methods.
>
> We further conducted ablation experiments of random projection (Figure 1, <https://anonymous.4open.science/r/7532-6C07/experiments2.pdf>), which confirm that using random projection in MICE does not reduce policy performance or increase constraint violations, while substantially reducing training time.
>
> Additionally, we examined the sensitivity of $N_k$ selection in KNN across multiple environments (Figure2, <https://anonymous.4open.science/r/7532-6C07/experiments2.pdf>). Results show that a larger  $N_k$ enhances policy safety by considering more unsafe states, but it increases computational overhead. Conversely, a smaller $N_k$ may not fully utilize memory information, potentially leading to higher constraint. To balance safety, performance, and efficiency, we set $N_k = 10$ in MICE for all tasks.
>
> We appreciate the reviewer’s constructive comments and hope that these analyses provide further clarity.
>
>
> >The baseline comparison with CPO (in Appendix C.3.1) should be moved to the main paper. I'd think that expanding to more tasks might verify that MICE is reliably safer than CPO.
>
> **Response:**
> We appreciate the reviewer’s valuable comments.
> Following your advice, we will move the comparison between MICE and CPO from Appendix to the main text for improved clarity.
>
> To further verify the safety advantages of MICE, we conducted additional experiments in a broader set of environments. The results in Figure 3 (<https://anonymous.4open.science/r/7532-6C07/experiments2.pdf>) demonstrate that MICE consistently achieves superior constraint satisfaction across all tasks, while CPO exhibits significant constraint violations.
>
> Additionally, to provide a more intuitive quantitative comparison, we present the difference in constraint violation rates between CPO and MICE during training. The results in Table 1 (<https://anonymous.4open.science/r/7532-6C07/experiments2.pdf>) show that CPO consistently exceeds MICE in violation rates in all environments, with an average violation rate 34.4\% higher than MICE. This further confirms the safety advantages of MICE.
>
>
> >The authors might want to consider expanding to other tasks to check how the discount rate of 0.99 behaves and comparing with PPO-Lagrangian and PID-Lagrangian.
>
> **Response:**
> We appreciate the reviewer’s valuable comment regarding the behaviour of discount factor 0.99 in more tasks and its comparison with PPO-Lagrangian and PID-Lagrangian.
> To address this, we evaluated MICE with a discount factor of 0.99 on additional tasks and compared its performance against PPO-Lagrangian and PID-Lagrangian. As shown in Figure 4 (<https://anonymous.4open.science/r/7532-6C07/experiments2.pdf>), MICE consistently achieves strong constraint satisfaction while maintaining superior policy performance. In contrast, PPO-Lagrangian and PID-Lagrangian exhibit significant oscillations in constraint during training.
>
> In our experiments, we used the consistent discount factors across all methods.
>
>
> [1] Hu, H., Ye, J., Zhu, G., Ren, Z., and Zhang, C. Generalizable episodic memory for deep reinforcement learning. In
> International Conference on Machine Learning, pp. 4380–4390. PMLR, 2021.
>
> [2] Zhu, G., Lin, Z., Yang, G., and Zhang, C. Episodic reinforcement learning with associative memory. In International
> Conference on Learning Representations.

---

> > ### Comment · Reviewer_rbBW · 2025-04-04
> >
> > I thank the authors for conducting additional experiments to clarify my doubts. I've raised my score as these points have been sufficiently addressed. Good luck!

---

> > > ### Author Response · Authors · 2025-04-04
> > >
> > > We sincerely appreciate your time and effort in reviewing our response and raising the score.

---

### Official Review · Reviewer_9DRS · 2025-03-16

**Overall Recommendation:** 4

**Summary:**

The paper tackles a important issue in CRL: the underestimation bias in the cost value function, arising from the functio approximation error, which often results in unsafe exploration and frequent constraint violations. The paper proposes the MICE algorithm to address this issue. It uses a flashbulb memory mechanism to store unsafe states and then compute an intrinsic cost based on the pseudo-count of state visits to high-risk regions.

**Claims And Evidence:**

The paper provides convergence proofs, worst-case constraint violation bounds, and other theoretical results that support its claims regarding bias correction and safe exploration.

**Essential References Not Discussed:**

No.

**Experimental Designs Or Analyses:**

The claim that the intrinsic cost mechanism alleviates underestimation without overly conservative behavior is supported by both theoretical analysis (via the adaptive balancing factor) and experimental results. However, the evidence might be less convincing if one considers potential sensitivity to hyperparameters or the handling of outlier states not captured by the flashbulb memory.

The use of KNN with random projection is theoretically motivated to reduce computation. However, the experiments do not extensively report on implementation details, computational overhead, or scalability issues, which might be relevant for real-world applications or more complex simulations.

**Methods And Evaluation Criteria:**

The proposed methods are well-aligned with the underestimation challenge inherent in  CRL.

**Other Comments Or Suggestions:**

1. Typically, for the KNN part, the choice of N_k is crucial. If N_k is too small, the count might be noisy; if it’s too large, it may smooth out important local variations. This parameter may require careful tuning depending on the environment. I suggest the authors to provide a hyperparamter sensitivity experiment on it.

2. Please provide the detailed architectural design, hyperparameter settings, or the exact implementation specifics of the random projection layer.

**Other Strengths And Weaknesses:**

1. The idea of using a flashbulb memory to derive an intrinsic cost is innovative and draws an interesting parallel with human risk perception.
2. The paper provides solid theoretical underpinnings including convergence analysis and constraint violation bounds. However, the assumptions (e.g., finite MDP, extensive sampling) might restrict practical applicability in more complex or continuous environments.

**Questions For Authors:**

I have a concern regarding the flashbulb memory mechanism described in the paper. The flashbulb memory stores only states where the extrinsic cost is greater than zero. This raises the issue that samples truly suffering from underestimation—possibly due to function approximation errors—may not be captured if they do not exhibit extrinsic cost > 0 or if they are outliers. Such outlier states are potentially more prone to underestimation, yet they may not fall within the types of states stored in the memory and therefore might not be adequately corrected by the intrinsic cost mechanism. Has the paper addressed this issue, and if so, what strategies are proposed to handle these cases?

**Relation To Broader Scientific Literature:**

The paper integrates and extends ideas from CRL, intrinsic motivation, and memory-based exploration. By combining these concepts, it provides a theoretically grounded and empirically validated framework

**Theoretical Claims:**

I reviewed the proofs.

---

> ### Author Rebuttal · Authors · 2025-03-31
>
> We sincerely appreciate the reviewer's positive and insightful comments. The following are the detailed responses to the points raised by Reviewer 9DRS.
>
> >The assumptions (e.g., finite MDP, extensive sampling) might restrict practical applicability in more complex or continuous environments.
>
> **Response:**
> We thank the reviewer’s valuable comment.
> To address this concern, we conducted additional experiments comparing MICE with CPO in SafetyCarButton1-v0 and SafetyPointButton1-v0 environments. These tasks are more complex, requiring agents to navigate to a target button and correctly press it while avoiding Gremlins and Hazards. Results presented in Figure 1 (<https://anonymous.4open.science/r/7532-6C07/experiments1.pdf>) show that MICE achieves superior constraint satisfaction while maintaining policy performance comparable to CPO.
>
> Furthermore, we acknowledge potential limitations where direct environmental sampling is infeasible. To address this, offline data can be leveraged to construct memory, thus enhancing MICE’s applicability in these special cases.
>
>
>
>
> >I suggest the authors provide a hyperparameter sensitivity experiment on $N_k$ in KNN.
>
> **Response:**
> We appreciate the reviewer’s valuable suggestion.
> To address this, we conducted hyperparameter sensitivity experiments for $N_k$ across multiple environments and analyzed the impact, as shown in Figure 2 in (<https://anonymous.4open.science/r/7532-6C07/experiments1.pdf>).
> The results show that increasing $N_k$ enhances the safety of the policy, as it considers more unsafe states in memory, but also raises computational overhead. Conversely, a smaller $N_k$ may lead to insufficient leveraging of unsafe state information, potentially leading to higher constraint. In this paper, we selected $N_k = 10$ uniformly across all environments to balance safety, performance, and computational efficiency.
>
>
>
> >Please provide the detailed implementation specifics of the random projection layer.
>
> **Response:**
> We appreciate the reviewer's valuable comment regarding the details of random projection.
>
> Our random projection layer is implemented using a Gaussian random matrix with shape $(n, m)$, projecting states from the original dimension $n$ to a lower embedding dimension $m$. According to the Johnson-Lindenstrauss lemma [1], this approach approximately preserves relative Euclidean distances in the original space, which is proven valid by prior KNN-based methods [2][3].
>
> In our work, this dimensionality reduction effectively decreases the computational complexity of KNN from $O(N n)$ to $O(N m)$, where $N$ is the number of states in memory. Specifically, in SafetyPointGoal1, the state dimension is reduced from 60 to 8.
>
> Furthermore, we conducted ablation experiment on random projection. As demonstrated in Figure 3 (<https://anonymous.4open.science/r/7532-6C07/experiments1.pdf>), utilizing random projection in MICE does not degrade policy performance or increase constraint violations, while effectively reducing training time.
>
>
>
> >The flashbulb memory only stores states with extrinsic cost greater than zero. However, the states with outlier extrinsic costs may not store in the memory and therefore might not be adequately corrected by the intrinsic cost mechanism. How does the paper handle these cases?
>
> **Response:**
> We appreciate the reviewer’s insightful question.
> In our tasks, the extrinsic cost is derived by the environment based on the agent's and obstacles' coordinates, providing unbiased real data without outliers.
> Underestimation primarily occurs due to constraint minimization during optimization, which disrupts the zero-mean property of noise, especially in regions with higher extrinsic costs. Therefore, current memory mechanism adequately captures these underestimated states in our tasks.
>
> However, we fully agree with the reviewer that in other special tasks, where extrinsic costs may be biased, outlier states could arise. To handle such cases, alternative criteria for adding new states into memory could be applied.
> One possible strategy is using the expected subsequent costs as the criterion for adding states into memory: a state would be stored if its expected future cost exceeds a threshold. This approach effectively mitigates the impact of individual outliers by considering multiple future states collectively.
>
> [1] Johnson, W. B., Lindenstrauss, J., et al. Extensions of lipschitz mappings into a hilbert space. Contemporary mathematics,
> 26(189-206):1, 1984.
>
> [2] Hu, H., Ye, J., Zhu, G., Ren, Z., and Zhang, C. Generalizable episodic memory for deep reinforcement learning. In
> International Conference on Machine Learning, pp. 4380–4390. PMLR, 2021.
>
> [3] Zhu, G., Lin, Z., Yang, G., and Zhang, C. Episodic reinforcement learning with associative memory. In International
> Conference on Learning Representations.

---

> > ### Comment · Reviewer_9DRS · 2025-04-03
> >
> > Thank you for the response. I appreciate your discussion regarding the corner case, and I look forward to seeing this discussion included in your revised version. I have raised my score accordingly.

---

> > > ### Author Response · Authors · 2025-04-03
> > >
> > > We sincerely appreciate you taking the time to review our response and raising the score. We will include these additional experiments and discussions into the revised version of the paper.

---

### Decision · Program_Chairs · 2025-05-01

**Decision:**

Accept (oral)

**Comment:**

All reviewers agreed that the work addresses an important problem that is understudied in the area of constrained reinforcement learning. Reviewers recognize that the proposed method is novel and that the theoretical claims are largely convincing. Reviewer 9DRS reviewed the proofs, but noted that some assumptions may affect the generality of some claims.
The rebuttal acknowledged the reviewer’s thoughtful feedback and provided detailed answers and clarifications, which sufficiently addressed the raised concerns.
I have no hesitation to strongly recommend this work for acceptance.